# RLIF: Interactive Imitation Learning as Reinforcement Learning

**Jianlan Luo**[*]   **Perry Dong**[*]   **Yuexiang Zhai**   **Yi Ma**   **Sergey Levine**
UC Berkeley; {jianlanluo, perrydong}@berkeley.edu

## Abstract

Although reinforcement learning methods offer a powerful framework for automatic skill acquisition, for practical learning-based control problems in domains such as robotics, imitation learning often provides a more convenient and accessible alternative. In particular, an interactive imitation learning method such as DAgger, which queries a near-optimal expert to intervene online to collect correction data for addressing the distributional shift challenges that afflict naïve behavioral cloning, can enjoy good performance both in theory and practice without requiring manually specified reward functions and other components of full reinforcement learning methods. In this paper, we explore how off-policy reinforcement learning can enable improved performance under assumptions that are similar but potentially even more practical than those of interactive imitation learning. Our proposed method uses reinforcement learning with user intervention signals *themselves* as rewards. This relaxes the assumption that intervening experts in interactive imitation learning should be near-optimal and enables the algorithm to learn behaviors that improve over the potential suboptimal human expert. We also provide a unified framework to analyze our RL method and DAgger; for which we present the asymptotic analysis of the suboptimal gap for both methods as well as the non-asymptotic sample complexity bound of our method. We then evaluate our method on challenging high-dimensional continuous control simulation benchmarks as well as real-world robotic vision-based manipulation tasks. The results show that it strongly outperforms DAgger-like approaches across the different tasks, especially when the intervening experts are suboptimal. Additional ablations also empirically verify the proposed theoretical justification that the performance of our method is associated with the choice of intervention model and suboptimality of the expert. Code and videos can be found on the project website: `rlif-page.github.io`

## 1 Introduction

Reinforcement learning methods have exhibited great success in domains where well-specified reward functions are available, such as optimal control, games, and aligning large language models (LLMs) with human preferences (Levine et al., 2016; Kalashnikov et al., 2018; Silver et al., 2017; Ouyang et al., 2022). However, imita-

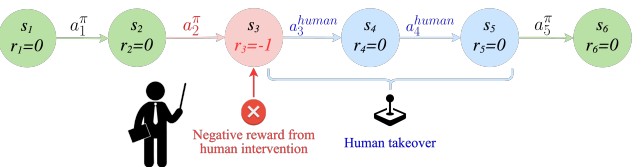

Figure 1: RLIF uses RL to learn without ground truth rewards, with data collected with suboptimal human interventions.

tion learning methods are still often preferred in some domains, such as robotics, because they are often more convenient, accessible, and easier to use. An often-cited weakness of naïve behavioral cloning is the compounding distributional shift induced by accumulating errors when deploying a learned policy. Interactive imitation learning methods, like the DAgger family of algorithms (Ross et al., 2011; Kelly et al., 2018; Hoque et al., 2022; Menda et al., 2018; Ross & Bagnell, 2014), address this issue by querying expert actions online and retraining the model iteratively in a supervised learning fashion. This performs well in practice, and in theory reduces the quadratic regret of imitation

---

[*]Equal contributions.

learning methods to be linear in the episode horizon. One particularly practical instantiation of this idea involves a human expert observing a learned policy, and intervening to provide *corrections* (short demonstrations) when the policy exhibits undesirable behavior (Kelly et al., 2018; Spencer et al., 2020). However, such interactive imitation learning methods still rely on interventions that are near-optimal, and offer no means to improve over the performance of the expert. Real human demonstrators are rarely optimal, and in domains such as robotics, teleoperation often does not afford the same degree of grace and dexterity as a highly tuned optimal controller. Can we combine the best parts of reinforcement learning and interactive imitation learning, combining the reward-maximizing behavior of RL, which can improve over the best available human behavior, with the accessible assumptions of interactive imitation learning?

The key insight we leverage in this work is that the decision to intervene during an interactive imitation episode itself can provide a reward signal for reinforcement learning, allowing us to instantiate RL methods that operate under similar but potentially weaker assumptions as interactive imitation methods, learning from human interventions but not assuming that such interventions are optimal. Intuitively, for many problems, it's easier to detect a mistake than it is to optimally correct it. Imagine an autonomous driving scenario with a safety driver. While the driver could intervene when the car deviates from good driving behavior, such interventions themselves might often be relatively uninformative and suboptimal – for example, if the human driver intervenes right before a collision by slamming on the breaks, simply teaching the policy to slam on the breaks is probably not the best solution, as it would be much better for the policy to learn to avoid the situations that necessitated such an intervention in the first place.

Motivated by this observation, we propose a method that runs RL on data collected from DAgger-style interventions, where a human operator observes the policy's behavior and intervenes with *suboptimal* corrections when the policy deviates from optimal behavior. Our method labels the action that leads to an intervention with a negative reward and then uses RL to minimize the occurrence of intervention by maximizing these reward signals. We call our method *RLIF*: Reinforcement Learning via Intervention Feedback. This offers a convenient mechanism to utilize non-expert interventions: the final performance of the policy would not be bottlenecked by the suboptimality of the intervening expert, but rather the policy would improve to more optimally avoid interventions happening at all. Of course, the particular intervention strategy influences the behavior of such a method, and we require some additional assumptions on when interventions occur. We formalize several such assumptions and evaluate their effect on performance, finding that several reasonable strategies for selecting *when* to intervene lead to good performance. We also provide a theoretical justification for the proposed method, via both an asymptotic analysis of the suboptimality gap that generalizes the theoretical framework of DAgger (Ross et al., 2011), and non-asymptotic analysis on learning an $\epsilon$-optimal policy with finite samples using the intervention rewards.

Our main contribution is a practical RL algorithm that can be used under assumptions that closely resemble interactive imitation learning, without requiring ground truth reward signals. We provide a theoretical analysis that studies under which conditions we expect this method to outperform DAgger-style interactive imitation techniques. Empirically, we evaluate our approach in comparison to DAgger on a variety of challenging continuous control tasks, such as the Adroit dexterous manipulation and Gym locomotion environments (Fu et al., 2020a). Our empirical results show that our method is on average **2-3x** better than best-performing DAgger variants, and this difference is much more pronounced as the suboptimality gap expands. We also demonstrate our method scales to a challenging real-world robotic task involving an actual human providing feedback. Our empirical results are well justified by our theory: suboptimal experts can in principle deteriorate the performance on both imitation learning and RL, but RL is generally more powerful than imitation learning. This is because RL can still recover the optimal policy $\pi^\star$ with additional samples, while imitation learning methods perform poorly due to the introduced suboptimality.

## 2 RELATED WORK

**Interactive imitation learning.** Imitation learning extracts policies from static offline datasets via supervised learning (Billard et al., 2008; Argall et al., 2009; Ho & Ermon, 2016; Osa et al., 2018; Laskey et al., 2017). Deploying such policies incurs distributional shift, because the states seen at deployment-time differ from those seen in training when the learned policy doesn't perfectly

match the expert, potentially leading to poor results (Ross & Bagnell, 2010; Ross et al., 2011). Interactive imitation learning leverages additional online human interventions from states visited by the learned policy to address this issue (Hoque et al., 2022; Kelly et al., 2018; Hoque et al., 2021; Menda et al., 2018; Ross et al., 2011). These methods generally assume that the expert interventions are near-optimal. Our method relaxes this assumption, by using RL to train on data collected in this interactive fashion, with rewards derived from the user's choice of when to intervene.

**Imitation learning with reinforcement learning.** Another line of related work uses RL to improve on suboptimal human demonstrations (Vecerik et al., 2017; Sun et al., 2017; Cheng et al., 2018; Sun et al., 2018; Nair et al., 2018; Ainsworth et al., 2019; Rajeswaran et al., 2018; Kidambi et al., 2020; Luo et al., 2021; Xie et al., 2022; Xue et al., 2023; Song et al., 2022; Ball et al., 2023). These methods typically initialize the RL replay buffer with human demonstrations, and then improve upon those human demonstrations by running RL with the task reward. In contrast to these methods, our approach does not require any task reward, but rather recovers a reward signal implicitly from intervention feedback. Some works use RL with interventions but assume the expert is optimal (Li et al., 2022b), which our method does not assume. Other works incorporate example high-reward states specified by a human user in place of demonstrations (Reddy et al., 2019; Eysenbach et al., 2021). While this is related to our approach of assigning negative rewards at intervention states, our interventions are collected interactively during execution under assumptions that match interactive imitation learning, rather than being provided up-front. Closely related to our approach, Kahn et al. (2020) proposed a robotic navigation system that incorporates *disengagement* feedback, where a model is trained to predict states where a user will halt the robot, and then avoids those states. Our framework is model-free and operates under general interactive imitation assumptions, and utilizes more standard DAgger-style interventions rather than just disengagement signals.

## 3   PRELIMINARIES AND PROBLEM SETUP

In this section, we set up the interactive imitation learning and RL formalism, and then introduce our problem statement.

**Behavioral cloning and interactive imitation learning.** The most basic form of imitation learning is behavioral cloning, which simply trains a policy $\hat{\pi}(a|s)$ on a dataset of demonstrations $D$, conventionally assumed to be produced by an optimal expert policy $\pi^\star(a|s)$, with $d_{\pi^\star}(s)$ as its state marginal distribution. Then, for each $(s, a) \in D$, behavioral cloning assumes that $s \sim d_{\pi^\star}(s)$ and $a \sim \pi^\star(a|s)$. Behavioral cloning then chooses $\hat{\pi} = \arg\min_{\pi \in \Pi} \sum_{s,a \in D} \ell(s, a, \pi)$, where $\ell(s, a, \pi)$ is some loss function, such as the negative log-likelihood (i.e., $\ell(s, a, \pi) = -\log \pi(a|s)$). Naïve behavioral cloning is known to accumulate regret quadratically in the time horizon $H$: when $\hat{\pi}$ differs from $\pi^\star$ even by a small amount, erroneous actions will lead to distributional shift in the visited states, which in turn will lead to larger errors (Ross et al., 2011). Interactive imitation learning methods, such as DAgger and its variants (Ross et al., 2011; Ross & Bagnell, 2010; Kelly et al., 2018; Hoque et al., 2022; Menda et al., 2018; Hoque et al., 2021), propose to address this problem, reducing the error to be *linear* in the horizon by gathering additional training data by running the learned policy $\hat{\pi}$, essentially adding new samples $(s, a)$ where $s \sim d_{\hat{\pi}}(s)$, and $a \sim \pi^\star(a|s)$. Different interactive imitation learning methods prescribe different strategies for adding such labels. Classic DAgger (Ross et al., 2011) runs $\hat{\pi}$ and then asks a human expert to relabel the resulting states with $a \sim \pi^\star(a|s)$. This is often unnatural in time-sensitive control settings, such as robotics

---

**Algorithm 1** Interactive imitation

**Require:** $\pi, \pi^{\text{exp}}, D$
1: **for** `trial` $i = 1$ to $N$ **do**
2:     Train $\pi$ on D via supervised learning
3:     **for** `timestep` $t = 1$ to $T$ **do**
4:         **if** $\pi^{\text{exp}}$ intervenes at $t$ **then**
5:             append $(s_t, a_t^{\pi^{\text{exp}}})$ to $D_i$
6:         **end if**
7:     **end for**
8:     $D \leftarrow D \cup D_i$
9: **end for**

---

and driving, and a more user-friendly alternative such as HG-DAgger and its variants (Kelly et al., 2018) instead allows a human expert to *intervene*, taking over control from $\hat{\pi}$ and overriding it with an expert action. We illustrate this in Algorithm 1.

Although this changes the state distribution, the essential idea of the method (and its regret bound) remain the same. However, as we will analyze further in Sec. 6, when the expert actions are *not* optimal (i.e., they come from a policy $\pi^{\text{exp}}$ that is somewhat worse than $\pi^\star$), the regret gap for DAgger-like methods expands. Our aim in this work will be to apply RL to this setting to address this issue, potentially even outperforming the expert.

**Reinforcement learning.** RL algorithms aim to learn optimal policies in Markov decision processes (MDPs). We will use an infinite-horizon formulation in our analysis. The MDP is defined as $\mathcal{M} = \{\mathcal{S}, \mathcal{A}, P, r, \gamma\}$. $\mathcal{M}$ comprises: $\mathcal{S}$, a state space of cardinality $S$, $\mathcal{A}$, an action space with size $A$, $P : \mathcal{S} \times \mathcal{A} \to \Delta(\mathcal{S})$, representing the transition probability of the MDP, $r : \mathcal{S} \times \mathcal{A} \to [0, 1]$ is the reward function, and $\gamma \in (0, 1)$ represents the discount factor. We use $\pi : \mathcal{S} \to \Delta(\mathcal{A})$. We introduce the value function $V^\pi(s)$ and the Q-function $Q^\pi(s, a)$ associated with policy $\pi$ as: $V^\pi(s) := \mathbb{E}\left[\sum_{t=0}^{\infty} \gamma^t r(s_t, a_t) | s_0 = s; \pi\right], \forall s \in \mathcal{S}$ and $\forall (s, a) \in \mathcal{S} \times \mathcal{A} : Q^\pi(s, a) := \mathbb{E}\left[\sum_{t=0}^{\infty} \gamma^t r(s_t, a_t) | s_0 = s, a_0 = a; \pi\right]$, as is standard in reinforcement learning analysis. We assume the initial state distribution is given by $\mu$: $s_0 \sim \mu$, and $\mu \in \Delta(\mathcal{S})$ and we slightly abuse the notation by using $V^\pi(\mu)$ to denote $\mathbb{E}_{s \sim \mu} V^\pi(s)$. The goal of RL is to learn an optimal policy $\pi^\star$ in the policy class $\Pi$ that maximizes the expected cumulative reward within the horizon $H$: $\pi^\star = \arg\max_{\pi \in \Pi} V^\pi(\mu)$ (Bertsekas, 2019). Without loss of generality, we assume the optimal policy $\pi^\star$ to be *deterministic* (Bertsekas, 2019; Li et al., 2022a). We slightly abuse the notation by using $V^\star, Q^\star$ to denote $V^{\pi^\star}, Q^{\pi^\star}$. Additionally, we use $d_\mu^\pi(s) = (1 - \gamma) \sum_{t=0}^{\infty} \gamma^t \mathbb{P}^\pi(s_t = s | s_0 \sim \mu)$, to denote the state occupancy distribution under policy $\pi$ on the initial state distribution $s_0 \sim \mu$. We also slightly abuse the notation by using $d_\mu^\pi \in \mathbb{R}^S$ to denote a vector, whose entries are $d_\mu^\pi(s)$.

**Problem setup.** Our aim will be to develop a reinforcement learning algorithm that operates under assumptions that resemble interactive imitation learning, where the algorithm is not provided with a reward function, but instead receives demonstrations followed by interactive interventions, as discussed above. We will not assume that the actions in the interventions themselves are optimal, but will make an additional mild assumption that the choice of *when* to intervene itself carries valuable information. We will discuss the specific assumption used in our analysis in Section 6, and we utilize several intervention strategies in our experiments, but intuitively we assume that the expert is more likely to intervene when $\hat{\pi}$ takes a bad action. This in principle can provide an RL algorithm with a signal to alter its behavior, as it suggests that the steps leading up to this intervention deviated significantly from optimal behavior. Thus, we will aim to relax the strong assumption that the expert is optimal in exchange for the more mild and arguably natural assumption that the expert's *choice of when to intervene* correlates with the suboptimality of the learned policy.

## 4 INTERACTIVE IMITATION LEARNING AS REINFORCEMENT LEARNING

The key observation of this paper is that, under typical interactive imitation learning settings, the *intervention signal alone* can provide useful information for RL to optimize against, without assuming that the expert demonstrator actually provides optimal actions.

Our method, which we refer to as RLIF (reinforcement learning from intervention feedback), follows a similar outline as Algorithm 1, but using reinforcement learning in place of supervised learning. The generic form of our method is provided in Algorithm 2. On Line 2, the policy is trained on the aggregated dataset $D$ using RL, with rewards derived from interventions. The reward is simply set to 0 for any transition where the expert does not intervene, and -1 for the previous transition where an expert intervenes. After an intervention, the expert can also optionally take over the control for a few steps;

---

**Algorithm 2** RLIF

**Require:** $\pi$, $\pi^{\text{exp}}$, $D$
1: **for** trial $i = 1$ to $N$ **do**
2:    Train $\pi$ on D via reinforcement learning.
3:    **for** timestep $t = 1$ to $T$ **do**
4:      **if** $\pi^{\text{exp}}$ intervenes at $t$ **then**
5:        label $(s_{t-1}, a_{t-1}, s_t)$ with -1 reward, append to $D_i$
6:      **else**
7:        label $(s_{t-1}, a_{t-1}, s_t)$ with 0 reward, append to $D_i$
8:      **end if**
9:    **end for**
10:   $D \leftarrow D \cup D_i$
11: **end for**

---

after which it will be released to the RL agent. This way, an expert can also intervene multiple times during one episode. All transitions are added to the dataset $D$, not just those that contain the expert reward; in practice, we can optionally initialize $D$ with a small amount of offline data to warm-start the process. An off-policy RL algorithm can utilize all of the data and can make use of the reward labels to avoid situations that cause interventions. This approach has a number of benefits: unlike RL, it doesn't require the true task reward to be specified, and unlike interactive imitation learning, it does not require the expert interventions to contain optimal actions, though it does require the *choice* of when to intervene to correlate with the suboptimality of the policy (as we will discuss later). Intuitively, we expect it to be less of a burden for experts to *only* point out which states are undesirable rather than actually *act optimally* in those states.

**Practical implementation.** To instantiate Algorithm 2 in a practical deep RL framework, it is important to choose the RL algorithm carefully. The data available for RL consists of a combination of on-policy samples and potentially suboptimal near-expert interventions, which necessitates using a suitable off-policy RL algorithm that can incorporate prior (near-expert) data easily but also can efficiently improve with online experience. While a variety of algorithms designed for online RL with offline data could be suitable (Song et al., 2022; Lee et al., 2022; Nakamoto et al., 2023), we adopt the recently proposed RLPD algorithm (Ball et al., 2023), which has shown compelling results on sample-efficient robotic learning. RLPD is an off-policy actor-critic reinforcement learning algorithm that builds on soft-actor critic (Haarnoja et al., 2018), but makes some key modifications to satisfy the desiderata above such as a high update-to-data ratio, layer-norm regularization during training, and using ensembles of value functions, which make it more suitable for incorporating offline data into online RL. For further details on this method, we refer readers to prior work (Ball et al., 2023), though we emphasize that our method is generic and in principle could be implemented relatively easily on top of a variety of RL algorithms. The RL algorithm itself is not actually modified, and our method can be viewed as a meta-algorithm that simply changes the dataset on which the RL algorithm operates.

## 5 EXPERIMENTS

Since our method operates under standard interactive imitation learning assumptions, our experiments aim to compare RLIF to DAgger under different types of suboptimal experts and intervention modes. We seek to answer the following questions: (1) Are the intervention rewards sufficient signal for RL to learn effective policies? (2) How well does our method perform compared to DAgger, especially with suboptimal experts? (3) What are the implications of different intervention strategies on empirical performance?

### 5.1 INTERVENTION STRATEGIES

In order to learn from interventions, we need the intervening experts to convey useful information about the task through their decision about *when* to intervene. Since real human experts are likely to be imperfect in making this decision, we study a variety of intervention strategies in simulation empirically to validate the stability of our method with respect to this assumption. The baseline strategy, which we call **Random Intervention**, simply intervenes uniformly at random, with equal probability at every step. The more intelligent model, **Value-Based Intervention**, strategy assumes the expert intervenes with probability $\beta$ when there is a gap $\delta$ between actions from the expert and agent w.r.t. a reference value function $Q^{\pi^{\text{ref}}}$.

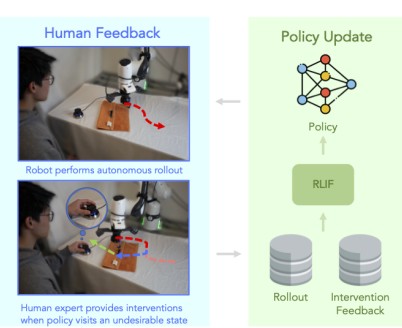

Figure 2: A human operator supervises policy training and provides intervention with a 3D mouse.

This model aims to capture an uncertain and stochastic expert, who might have a particular policy $\pi^{\text{exp}}$ that they use to choose actions (which could be highly suboptimal), and a *separate* value function $Q^{\pi^{\text{ref}}}$ with its corresponding policy $\pi^{\text{ref}}$ that they use to determine if the robot is doing well or not, which they use to determine when to intervene. Note that $\pi^{\text{ref}}$ might be much better than $\pi^{\text{exp}}$ – for example, a human expert might correctly determine the robot is choosing poor actions for the

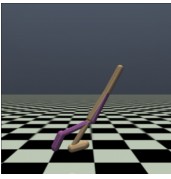 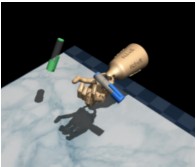 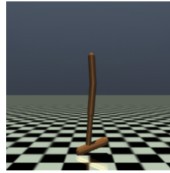 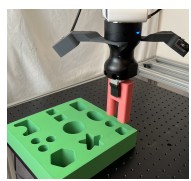 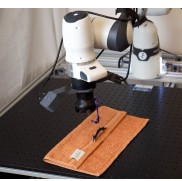

Figure 4: **Tasks in our experimental evaluation**: Benchmark tasks Walker2d, Pen, and Hopper and two vision-based contact-rich manipulation tasks on a real robot. The benchmark tasks require handling complex high-dimensional dynamics and underactuation. The robotic insertion task requires additionally addressing complex inputs such as images, non-differentiable dynamics such as contact, and all sensor noise associated with real-world robotic settings.

task, even if their own policy $\pi^{\text{exp}}$ is not good enough to perform the task either. $\delta$ represents the confidence level of the intervening expert: the smaller $\delta$ is, the more willing they are to intervene on even slightly suboptimal robot actions. We formalize this model in Eq. 5.1. In practice, we choose a value for $\beta$ close to 1, such as 0.95.

$$\mathbb{P}(\textit{Intervention}|s) = \begin{cases} \beta, & \text{if } Q^{\pi^{\text{ref}}}(s, \pi^{\text{exp}}(s)) > Q^{\pi^{\text{ref}}}(s, \pi(s)) + \delta \\ 1 - \beta, & \text{otherwise.} \end{cases} \tag{5.1}$$

This model may not be fully representative of real human behavior, so we also evaluate our method with real human interventions, where an expert user provides intervention feedback to a real-world robotic system performing a peg insertion task, shown in Figure 2. We discuss this further in Sec. 5.3.

In Appendix A.1, we also present a didactic experiment with the Grid World environment where we use value iteration as the underlying RL algorithm, to verify that RLIF does indeed converge to the optimal policy under idealized assumptions when we remove approximation and sampling error of the RL method from consideration.

## 5.2 PERFORMANCE ON CONTINUOUS CONTROL BENCHMARK TASKS

First, we evaluate RLIF in comparison with various interactive imitation learning methods on several high-dimensional continuous control benchmark tasks. These experiments vary both the optimality of the expert's policies and the expert's intervention strategy.

**Simulation experiment setup.** We use Gym locomotion and Adroit dexterous manipulation tasks in these experiments, based on the D4RL environments (Fu et al., 2020b). The Adroit environments require controlling a 24-DoF robotic hand to perform tasks such as opening a door or rotating a pen to randomly sampled goal locations. The Gym locomotion task (walker) requires controlling a planar 2-legged robot to walk forward. Both domains require continuous control at relatively high frequencies, where approximation errors and distributional shift can accumulate quickly for naïve imitation learning

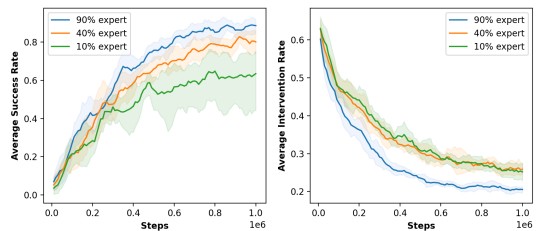

Figure 3: Average success rate and intervention rate for the Adroit-Pen task during training, as the agent improves, the intervention decreases.

methods. Note that although these benchmarks are often used to evaluate the performance of RL algorithms, we do not assume access to any reward function beyond the signal obtained from the expert's interventions, and therefore our main point of comparison are DAgger variants, rather than prior RL algorithms.

The experts and $Q^{\pi^{\text{ref}}}$ associated with intervention strategies are obtained by training policies on subsampled datasets to induce the desired level of suboptimality. Further details on initial policy and intervention expert training can be found in Appendix. A.3.

**Results and discussion.** We report our main results in Table. 4, with additional learning curves for analyzing learning progress in terms of the number of iterations presented in Appendix. A.5 and an IQL baseline in Appendix. A.7. Each table depicts different expert performance levels on different tasks (rows), and different methods with different intervention strategies (columns).

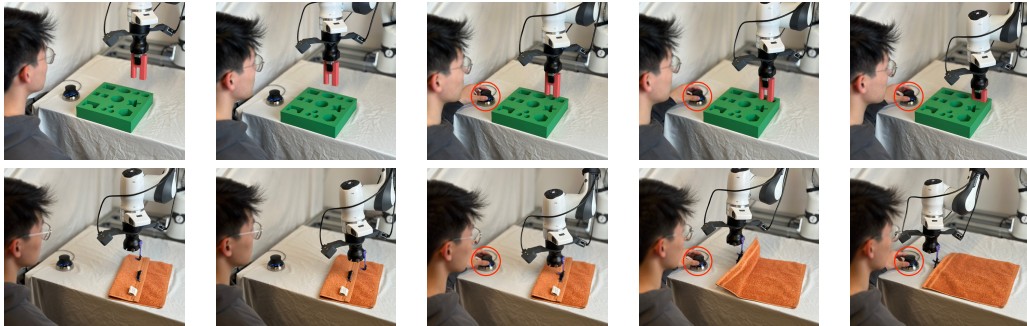

Figure 5: Sequential steps of robot manipulation for the peg insertion and cloth unfolding tasks on a real robot.

| Domain | Expert Level | RLIF with Value Based Intervention | RLIF with Random Intervention | HG-DAgger | HG-DAgger with 85% Random Intervention | DAgger | DAgger with 85% Random Intervention | BC |
|---|---|---|---|---|---|---|---|---|
| adroit-pen | ~90% | **88.47±3.06** | 42.87±12.86 | 73.47±6.19 | 74.27±5.79 | 78.13±3.24 | 79.07±9 | |
| | ~40% | **80.87±6.01** | 34.13±10.32 | 60±3.58 | 29.33±6.56 | 35.73±7.49 | 38.67±2.06 | 54.13±14.24 |
| | ~10% | **64.04±17.59** | 28.33±4.43 | 28.53±7.66 | 9.47±4.09 | 8.93±1.43 | 12.8±6.25 | |
| | average | **77.79±8.89** | 35.11±9.20 | 54±5.81 | 37.69±5.48 | 40.93±4.05 | 43.51±5.77 | 54.13±14.24 |
| locomotion-walker2d | ~110% | 108.99±5.28 | 106.51±0.47 | 53.55±9.76 | **112.7±2.51** | 57.94±8.69 | 76.13±3.27 | |
| | ~70% | **99.66±5.9** | 75.62±50.02 | 44.75±3.46 | 69.73±5.99 | 20.49±3.15 | 43.59±2.56 | 44.46±13.59 |
| | ~20% | **102.85±2.26** | 19.11±24.08 | 11.94±0.88 | 19.66±3.69 | 12.37±2.96 | 20.1±2.17 | |
| | average | **103.83±4.48** | 67.08±43.83 | 36.75±4.7 | 67.36±4.06 | 30.27±4.93 | 46.61±2.67 | 44.46±13.59 |
| locomotion-hopper | ~110% | **109.17±0.16** | 93.76±7.87 | 80.3±14.74 | 86.93±4.85 | 70.58±9.98 | 61.64±11.36 | |
| | ~40% | **108.42±0.62** | 103.9±10.28 | 40.66±3.35 | 42.65±2.86 | 38.7±3.7 | 19.63±2.3 | 64.77±10.23 |
| | ~15% | **108.01±0.64** | 75.12±28.95 | 25.2±3.58 | 24.37±2.26 | 19.54±2.14 | 10.29±1.24 | |
| | average | **108.53±0.47** | 90.93±15.7 | 48.72±7.22 | 51.32±3.32 | 42.94±5.27 | 30.46±4.97 | 64.77±10.23 |

Table 1: A comparison of RLIF, HG-DAgger, DAgger, and BC on continuous control tasks. RLIF consistently performs better than HG-DAgger and DAgger baselines for each individual expert level as well as averaged over all expert levels.

Note the "expert level" in the table indicates the suboptimality of the expert; for example, a "90%" expert means it can achieve 90% of the reference optimal expert score of a particular task. This score is normalized across environments to be 100.0. Since we trained such experts using our curated datasets, the normalized score can exceed 100.0. For the rest of the table, we report the performance of different methods w.r.t. this normalized reference score.

To start the learning process, we initialize the replay buffer with a small number of samples. The details of the datasets can be found in Appendix. A. We compare RLIF with DAgger and HG-DAgger under different intervention modes and expert levels. Specifically, we run DAgger under two intervention modes: 1) the expert issues an intervention if the difference between the agent's action and the expert's action is larger than a threshold, as used in Kelly et al. (2018); Ross et al. (2011), 2) the expert issues

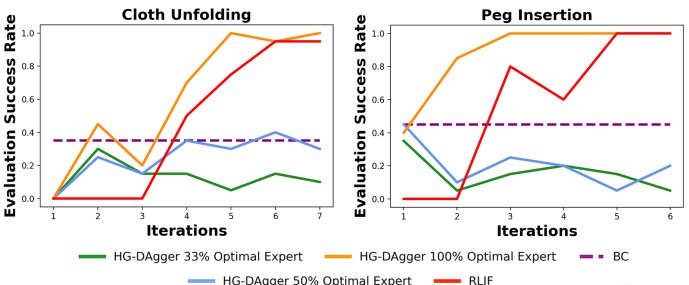

Figure 6: RLIF on the real-world robotic manipulation task.

random intervention uniformly with a given probability at each step of the episode. The results suggest several takeaways: (1) We see that regardless of the suboptimality of the experts, RLIF with value-based interventions can indeed reach good performance, even if the suboptimality gap is *very large*, such as improving a 15% expert to a score over 100.0 in the 2D-walker task. (2) RLIF with a value-based simulated interventions outperforms RLIF with random interventions, especially when the suboptimality is large. This confirms that the interventions carry a meaningful signal about the task, providing implicit rewards to RLIF. We also observe that the value-based intervention rate goes down as the agent improves, as shown in Fig. 3. This confirms our proposed intervention model does carry useful information about the task, and works reasonably as the agent learns. (3) RLIF outperforms both DAgger and HG-DAgger consistently across all tasks, and the performance gap is more pronounced when the expert is *more suboptimal*, and can be as large as **5x**; crucially, this resonates with our motivation: the performance of DAgger-like algorithms will be subject to the

suboptimality of the experts, while our method can reach good performance even with suboptimal experts by learning from the expert's decision of *when* to intervene.

**Ablations on Intervention Modes.** As stated in Sec. 5.1, the performance of RLIF critically depends on when the expert chooses to intervene. Now we analyze the effect of different $Q^{\pi^{\text{ref}}}$ value functions on the final performance of RLIF. We report the numbers in Table. 3 of Appendix. A.6. We can observe that the particular choice of $Q^{\pi^{\text{ref}}}$ does heavily influence our algorithm's performance: as the $\pi^{\text{ref}}$ deteriorates, $Q^{\pi^{\text{ref}}}$ becomes increasingly inaccurate, making the intervention decision more "uncalibrated." This translates to worse policy performance.

## 5.3 REAL-WORLD VISION-BASED ROBOTIC MANIPULATION TASK

While we have shown that RLIF works well under reasonable models of the expert's intervention strategy, to confirm that this model actually reflects real human interventions, we next conduct an experiment where a human operator supplies the interventions directly for a real-world robotic manipulation task. The first task, shown in Fig. 2, involves controlling a 7-DoF robot arm to fit a peg into its matching shape with a very tight tolerance (1.5mm), directly from image observations. This task is difficult because it involves dealing with discontinuous and non-differentiable contact dynamics, complex high-dimensional inputs from the camera, and an imperfect human operator. The second task involves controlling the same robot arm to unfold a piece of cloth by hooking onto it and pulling it open. This task is difficult because it requires manipulation of a deformable object: specifying rewards programmatically for such a task is very challenging, and the perception system trained via RL must be able to keep track of the complex object geometry. Filmstrips of the tasks are shown in Fig. 5 to visualize the sequential steps of both the peg insertion and cloth unfolding tasks. More details on the setup can be found in Appendix. A.2. We report the results in Fig. 6. Our method can solve the insertion task with a 100% success rate within six rounds of interactions, which corresponds to 20 minutes in wall-clock time, including robot resets, computation, and all intended stops. It solves the unfolding task with a 95% success rate in seven rounds of interaction. This highlights the practical usability of our method in challenging real-world robotic tasks. We refer the training and evaluation videos of the robotic tasks to our website: `rlifpaper.github.io`

## 6 THEORETICAL ANALYSIS

In this section, we provide a theoretical analysis of RLIF in terms of its suboptimality gap. We introduce our theoretical settings and goals in Sec. 6.1, and quantify the suboptimality gap in Sec. 6.2.

### 6.1 THEORETICAL SETTINGS AND ASSUMPTIONS

We consider the setting where we cannot access to the true task reward $r$, and we can only obtain rewards $\tilde{r}_\delta$ through interventions. For simplicity of analysis, we define this reward function as $\tilde{r}_\delta = \mathbf{1}\{\text{Not intervened}\}$ (i.e., 1 for each step when an intervention does *not* happen), which differs from the reward in Algorithm 2 by a constant and thus does not change the result, but allows us to keep rewards in the range $[0, 1]$. We aim to apply RL using the intervention reward $\tilde{r}_\delta$ induced by RLIF, and quantify the suboptimality gap with respect to the true reward $r$. In particular, let $\tilde{\pi}$ denote the policy learned by RLIF for maximizing the overall return on reward function $\tilde{r}_\delta$:

$$\tilde{\pi} \in \Pi_\delta^{\text{opt}}, \text{ s.t. } \Pi_\delta^{\text{opt}} := \arg\max_{\pi \in \Pi} \mathbb{E}_{a_t \sim \pi(s_t), \tilde{r}_\delta(s_t, \pi(s_t))} \left[ \sum_{t=0}^{\infty} \gamma^t \tilde{r}_\delta(s_t, a_t) | s_0 \sim \mu \right]. \quad (6.1)$$

Our goal is to bound the the suboptimality gap: $\text{SubOpt} := V^\star(\mu) - V^{\tilde{\pi}}(\mu)$. Based on the intervention strategy specified in Eqn. 5.1, we introduce our definitions of $\tilde{r}_\delta$ as follows. We leave the discussion of the property of the function space $\Pi_\delta^{\text{opt}}$ to Appendix E.

**Assumption 6.1.** *For a policy $\pi \in \Pi$, the reward function $\tilde{r}_\delta$ induced by Alg. 2 is defined as $\tilde{r}_\delta = \mathbf{1}\{\text{Not intervened}\}$. The probability of intervention is similarly defined as Eqn. 5.1, for a constant $\beta > 0.5$. We assume the expert may refer to a reference policy $\pi^{\text{ref}} \in \Pi_\delta^{\text{opt}}$ for determining intervention strategies as mentioned in Sec. 5.1. In addition, we assume the expert's own policy will not be intervened $\pi^{\text{exp}} \in \Pi_\delta^{\text{opt}}$.*

The $\delta$ that appears in $\tilde{r}_\delta$ for determining the intervention condition is similarly defined in Eqn. 5.1 could be referred as the "confidence level" of the expert. Note that Assumption 6.1 is a generalized version of the DAgger, as in the DAgger case one can treat $\pi^{\text{ref}} = \pi^{\text{exp}}$.

## 6.2 MAIN RESULT

To provide a comparable suboptimality gap with DAgger (Ross et al., 2011), we introduce the similar behavioral cloning loss on $\pi^{\text{exp}}$.

**Definition 6.2** (Behavior Cloning Loss (Ross et al., 2011))**.** *We use the following notation to denote the 0-1 loss on behavior cloning loss w.r.t. $\pi^{\text{exp}}$ and $\pi^{\text{ref}}$ as:* $\ell(s, \pi(s)) = \mathbf{1}\left\{\pi(s) \neq \pi^{\text{exp}}(s)\right\}, \forall s \in \mathcal{S}$, $\ell'(s, \pi(s)) = \mathbf{1}\left\{\pi(s) \neq \pi^{\text{ref}}(s)\right\}, \forall s \in \mathcal{S}$.

With Assumption 6.1, and Def. 6.2, we now introduce the suboptimality gap of RLIF.

**Theorem 6.3** (Suboptimality Gap of RLIF)**.** *Let $\tilde{\pi} \in \Pi_\delta^{\text{opt}}$ denote an optimal policy from maximizing the reward function $\tilde{r}_\delta$ generated by RLIF. Let $\epsilon = \max\left\{\mathbb{E}_{s \sim d_\mu^{\tilde{\pi}}} \ell(s, \pi(s)), \mathbb{E}_{s \sim d_\mu^{\tilde{\pi}}} \ell'(s, \pi(s))\right\}$ (Def. 6.2). Under Assumption 6.1, when $V^{\pi^{\text{ref}}}$ is known, the RLIF suboptimality gap satisfies:*

$$\text{SubOpt}_{\text{RLIF}} = V^\star(\mu) - V^{\tilde{\pi}}(\mu) \leq \min\left\{V^{\pi^\star}(\mu) - V^{\pi^{\text{ref}}}(\mu), V^{\pi^\star}(\mu) - V^{\pi^{\text{exp}}}(\mu)\right\} + \frac{\delta\epsilon}{1 - \gamma}.$$

Note that our Assumption 6.1 is a generalized version of DAgger with the additional $V^{\pi^{\text{ref}}}$, we can use the same analysis framework of Thm. 6.3 to obtain a suboptimality gap of DAgger and obtain the following suboptimality gap. Using a similar proof strategy to Thm. 6.3, we provide a suboptimality gap for DAgger as follows.

**Corollary 6.4** (Suboptimality Gap of DAgger)**.** *Let $\tilde{\pi} \in \Pi_\delta^{\text{opt}}$ denote an optimal policy from maximizing the reward function $\tilde{r}_\delta$ generated by RLIF. Let $\epsilon = \mathbb{E}_{s \sim d_\mu^{\tilde{\pi}}} \ell(s, \pi(s))$ (Def. 6.2). Under Assumption 6.1, when $V^{\pi^{\text{ref}}}$ is unknown, the DAgger suboptimality gap satisfies:*

$$\text{SubOpt}_{\text{DAgger}} = V^\star(\mu) - V^{\tilde{\pi}}(\mu) \leq V^{\pi^\star}(\mu) - V^{\pi^{\text{exp}}}(\mu) + \frac{\delta\epsilon}{1 - \gamma}. \tag{6.2}$$

A direct implication of Thm. 6.3 and Cor. 6.4 is that, under Assumption 6.1, RLIF is at least as good as DAgger, since $\text{SubOpt}_{\text{RLIF}} \leq \text{SubOpt}_{\text{DAgger}}$.

We provide a bandit example in Appendix. B.3 to show that our upper bounds in Thm. 6.3 and Cor. 6.4 are tight. We leave the proof of Thm. 6.3 and Cor. 6.4 analysis under our framework to Appendix B. For completeness, we also provide a non-asymptotic sample complexity analysis for learning an $\hat{\pi}$ that maximizes $V_{\tilde{r}_\delta}^\pi(\mu)$ in Appendix C. Our non-asymptotic result in Cor. C.3 suggests that the total sample complexity *does not exceed* the sample complexity $\widetilde{O}\left(\frac{SC_{\text{exp}}^\star}{(1-\gamma)^3\epsilon^2}\right)$, where $C_{\text{exp}}^\star$ is the concentrability coefficient w.r.t. $\pi^{\text{exp}}$ (formally defined in Def. C.1).

## 7 DISCUSSION AND LIMITATIONS

We presented a reinforcement learning method that learns from interventions in a setting that closely resembles interactive imitation learning. Unlike conventional imitation learning methods, our approach does not rely strongly on access to an optimal expert, and unlike conventional reinforcement learning algorithms, it does not require access to the ground truth reward function, instead deriving a reward signal from the expert's decision about when to intervene. However, our approach does have a number of limitations. First, we require an RL method that can actually train on all of the data collected by both the policy and the intervening expert. This is not necessarily an easy RL problem, as it combines off-policy and on-policy data. That said, we were able to use an off-the-shelf offline RL algorithm (RLPD) without modification. Second, imitation learning is often preferred precisely because it doesn't require online deployment at all, and this benefit is somewhat undermined by interactive imitation learning methods. While in practice deploying a policy under expert oversight might still be safer (e.g., with a safety driver in the case of autonomous driving), investigating the safety challenges with such online exploration is an important direction for future work.

## 8 ACKNOWLEDGEMENTS

This research was partly supported by ARO W911NF-21-1-0097, and ONR N00014-21-1-2838 and N00014-20-1-2383, the Berkeley Research Computing Platform (BRC), NSF Cloud Bank program.

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

## A  ADDITIONAL EXPERIMENT RESULTS AND DETAILS

### A.1  GRID WORLD NAVIGATION AS A DIAGNOSTIC TASK

We start with a didactic task in a Grid World environment to visualize the behavior of our method over the course of training. The MDP is a 6x6 grid, where the task is to navigate from location $(1, 1)$ to $(6, 6)$. The (unobserved) optimal policy is produced by optimizing a reward that is sampled from $[-0.1, 0]$ at each point on an optimal route, and from $[-1, -0.1]$ for any state that is not on this route, such that a good policy must follow the route precisely. We employ the **Value-Based Intervention** strategy. We collect five intervention trajectories per round following this strategy. We use value iteration as the RL algorithm since this MDP can be computed exactly. We plot the results in Fig. 7, visualizing the value function as well as the actions learned by our method. As the expert applies interventions on successive iterations, RLIF determines a reasonable value function, with high (red) values along the desired route and low (blue) values elsewhere, arrow indicating the action actions; despite not actually observing the true task reward. We also plot the true value function of this task in Fig. 7(f). While of course the true value function differs from the solution found by RLIF, the policy matches on the path from the start to the goal, indicating that RLIF can successfully learn the policy from only intervention feedback when we abstract away sampling error.

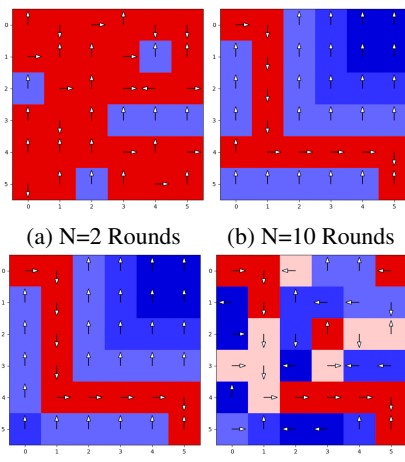

(a) N=2 Rounds    (b) N=10 Rounds

(c) N=20 Rounds    (d) True task values

Figure 7: Learned value function at successive rounds of RLIF (red is higher). Over the course of the algorithm, the value function and policy converge on the optimal path through the grid world, only using intervention-based rewards.

### A.2  ROBOT EXPERIMENT DETAILS

**Task Description.**   For real robot experiments, we perform the task of peg insertion into 3D-printed board and cloth unfolding with velcro hooks using a 7-DoF Franka Research 3 robot arm. The RL agent stream control commands to the robot at 10 HZ, each episode is at a maximum of 50 timesteps, which converts to 5 seconds. The robot obtains visual feedback from the Intel Realsense D405 cameras mounted on its end-effectors. In our setup, the human operator provides interventions by using a 3D mouse. Whenever the operator touches the mouse, they take over control of the robot, and can return control to the RL agent by disengaging with the mouse. We mark the transition of the previous time step of such an intervention as a negative reward. We use an ImageNet pre-trained EfficientNet-B3 (Tan & Le, 2019) as a vision backbone for faster policy training. Two cameras mounted on the robot end-effector provide continuous visual feedback. For the insertion task, a trial is counted as a success if the peg is inserted into its matching hole with a certain tolerance, and for the unfolding task, a trial is counted as a success if the cloth is successfully unfolded all the way. All success rates are reported based on 20 trials.

**Collecting Trajectories.**   For RLIF, DAgger-based baselines, and BC, five suboptimal trajectories are used to initialize the replay buffer. During training, human interventions are given when the policy appears to be performing suboptimally. The BC results are trained on the same five trajectories that initialize the RLIF buffer.

### A.3  EXPERIMENT SETUP DETAILS

**Success Rates.**   For the Adroit tasks, success rate is used as a measure of performance since whether the agent learns a policy that can success in the task can be more informative of how well expert interventions are incorporated than reward. The evaluation is done on the Gymnasium sparse environments AdroitHandPenSparse-v1 and AdroitHandDoorSparse-v1, where a trajectory is determined to be a success if at the end the agent receives a reward of 10 corresponding to success in the Gymnasium environments. Walker2d and Hopper uses average normalized returns since success rate is hard to definitively measure for the locomotion environments.

**Offline Datasets.** To perform controlled experimentation of RLIF against DAgger and HG-DAgger, we prepare offline datasets to use for pretraining on DAgger and HG-DAgger and to initialize to replay buffer for RLIF. The initial datasets of all simulation tasks are subsets of datasets provided in d4rl. The specific dataset used to subsample and sizes of the initial dataset for each task are listed in Table 2. The rewards for all timesteps for all of the initial datasets are set to zero.

| Tasks | Dataset | Subsampled Size |
|---|---|---|
| Adroit Pen | pen-expert-v1 | 50 trajectories |
| Locomotion Hopper | hopper-expert-v2 | 50 trajectories |
| Locomotion Walker2d | walker2d-expert-v2 | 10 trajectories |
| Real-World Robot Experiments | manually collected suboptimal trajectories | 5 trajectories |

Table 2: Offline datasets for each task.

**Expert Training.** Experts of varying levels are trained for all tasks on the dataset with either BC, IQL, SAC, or RLPD depending on the task. The various levels of the experts are obtained by training on the human dataset or expert datasets subsampled to various sizes. For each level and task, the same expert is used to intervene across all intervention strategies for both RLIF, DAgger, and HG-DAgger.

## A.4 INTERVENTION STRATEGIES

**Random Intervention.** For the random intervention strategy, in our experiments interventions are sampled uniformly at random. We consider 30%, 50%, and 85% intervention rates, where the intervention rate is computed as the number of steps under intervention over the total number of steps. We define $i$ as the particular timestep where an expert choose to intervene, and $k$ as the number of steps an expert takes over after such an intervention. At any given timestep $t$ that is not intervened by the expert, we can then define the uniformly random intervention strategy with different probabilities as below:

30% Intervention Rate: $I \sim \mathcal{U}(t+1, t+10)$ and $k \sim \mathcal{U}(1, 5)$

50% Intervention Rate: $I \sim \mathcal{U}(t+1, t+5)$ and $k \sim \mathcal{U}(3, 7)$

85% intervention Rate: $I \sim \mathcal{U}(t+1, t+2)$ and $k \sim \mathcal{U}(12, 16)$.

**Value-based Intervention.** For value-based intervention strategy, we compare the value of the current state and policy actions to the value of the current state and expert actions under the reference $Q^{\pi^{\mathrm{ref}}}$ function.

Specifically, we follow the intervention strategy below where an intervention is likely to occur if the expert values are greater than the policy values by a threshold.

$$\mathbb{P}(Intervention|s) = \begin{cases} \beta, & \text{if } Q^{\pi^{\mathrm{ref}}}(s, \pi^{\exp}(s)) > Q^{\pi^{\mathrm{ref}}}(s, \pi(s)) + \delta \\ 1 - \beta, & \text{otherwise.} \end{cases} \tag{A.1}$$

In practice, we found a relative threshold comparison effective, which is stated as below:

$$\mathbb{P}(Intervention|s) = \begin{cases} \beta, & \text{if } Q^{\pi^{\mathrm{ref}}}(s, \pi^{\exp}(s)) * \alpha > Q^{\pi^{\mathrm{ref}}}(s, \pi(s)) \\ 1 - \beta, & \text{otherwise.} \end{cases} \tag{A.2}$$

We choose a value for $\beta$ close to 1 such as 0.95 and a value of $\alpha$ close to 1 such as 0.97.

## A.5 EXPERIMENT PLOTS

We present some representative training plots below to describe the learning process of RLIF and HG-DAgger. The plots cover a variety of expert levels and intervention strategies.

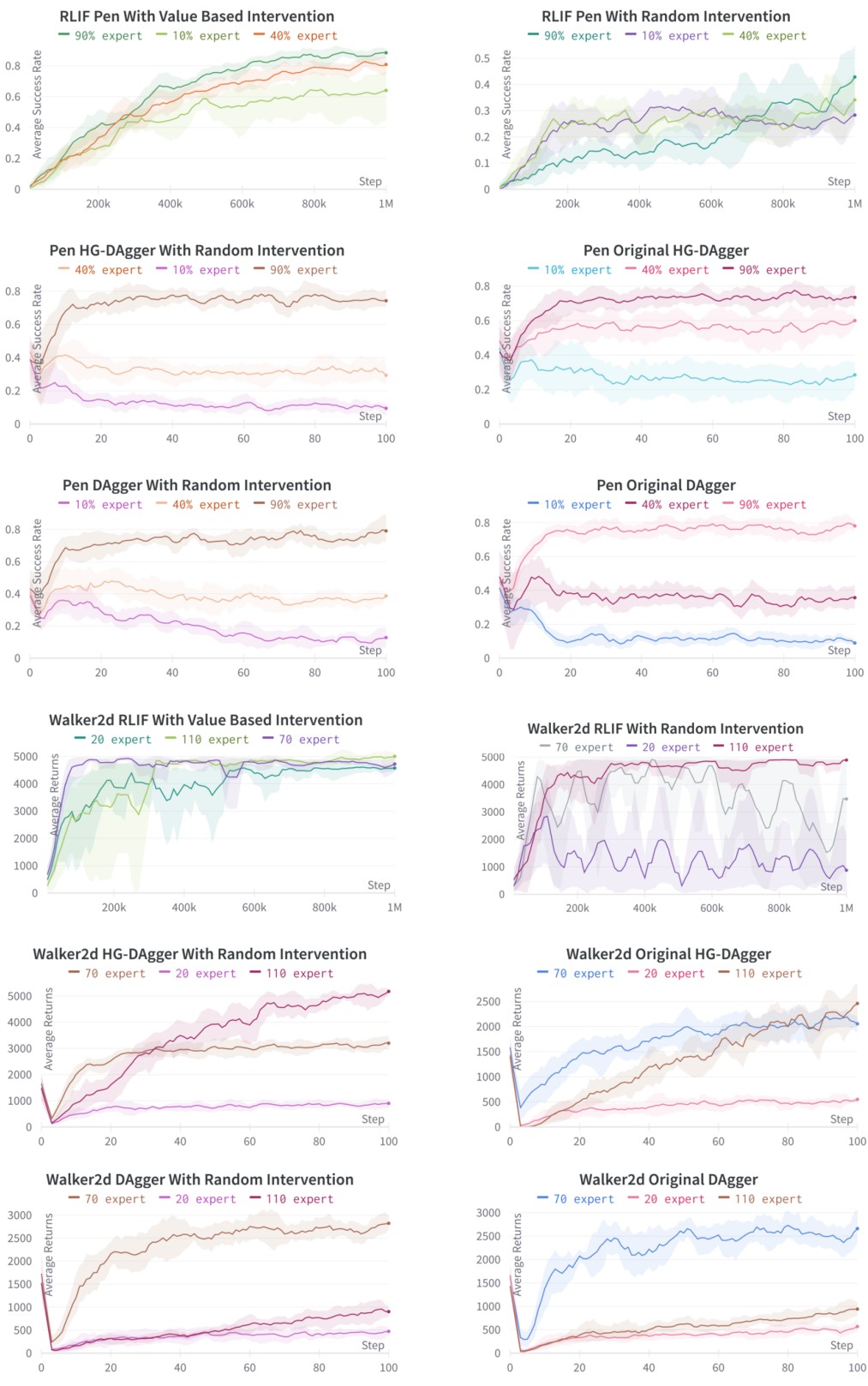

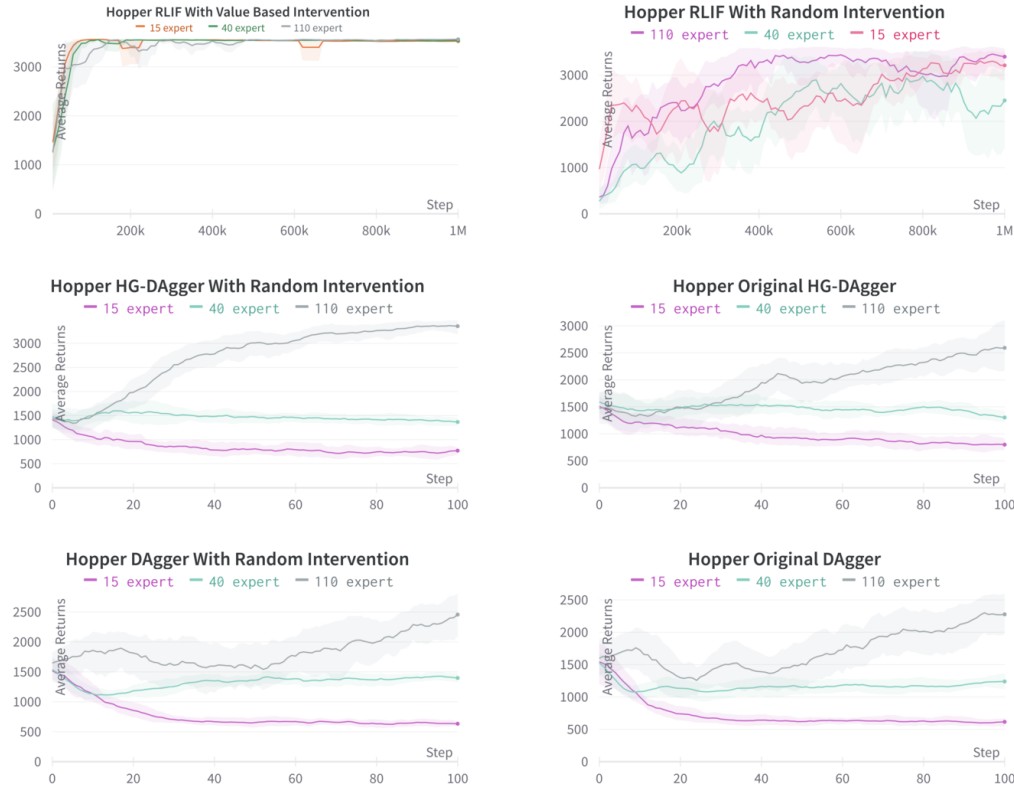

Figure 9: Training Plots of RLIF, DAgger, and HG-DAgger

## A.6 Ablation Experiment Results

| Domain | Expert Level | 10% $Q^{\pi^{\mathrm{ref}}}$ | 60% $Q^{\pi^{\mathrm{ref}}}$ | 110% $Q^{\pi^{\mathrm{ref}}}$ |
|---|---|---|---|---|
| locomotion -walker | ~110% | 78.9±49.97 | 85.91±19.31 | 108.99±5.28 |
| | ~40% | 38.78±20.3 | 71.21±21.96 | 99.66±5.9 |
| | ~20% | 33.55±9.63 | 66.8±8.22 | 102.85±2.26 |

Table 3: An ablation of RLIF on walker2d with different $Q^{\pi^{\mathrm{ref}}}$ and expert levels. A 10% $Q^{\pi^{\mathrm{ref}}}$ means that it was trained with the offline dataset generated by a 10% expert.

## A.7 Additional Baselines

| Domain | Expert Level | IQL with Value Based Intervention | IQL with Random Intervention |
|---|---|---|---|
| adroit-pen | ~90% | 42.5±2.26 | 68.83±3.75 |
| | ~40% | 46.58±2.81 | 25.92±3.52 |
| | ~10% | 40.67±5.16 | 12.08±4.39 |
| | average | 43.25±3.41 | 35.61±3.89 |
| locomotion-walker2d | ~110% | 20.37±5.48 | 80.22±12.98 |
| | ~70% | 47.97±14.94 | 56.33±7.45 |
| | ~20% | 47.02±4.83 | 14.7±1.92 |
| | average | 38.45±8.42 | 50.42 ±7.45 |
| locomotion-hopper | ~110% | 26.37±2.04 | 37.91±3.48 |
| | ~40% | 28.98±4.11 | 27.88±1.84 |
| | ~15% | 25.32±4.81 | 16.64±1.8 |
| | average | 26.89±3.65 | 27.48±2.37 |

Table 4: A comparison of RLIF, HG-DAgger, DAgger, and BC on continuous control tasks. RLIF consistently performs better than HG-DAgger and DAgger baselines for each individual expert level as well as averaged over all expert levels.

## A.8 EXPERIMENT HYPERPARAMETERS

**Training Parameters.** We set the number of rounds to $N = 100$ and the number of trajectories collected per round to 5. We also use the number of pretraining epochs and pretraining train steps per epoch to 200 and 300, and the epochs and train steps per epoch for each round to 25 and 100 to achieve consistent training.

| Tasks | Parameters | Values |
|---|---|---|
| Adroit Pen | UTD Ratio | 5 |
| Locomotion Hopper | UTD Ratio | 15 |
| Locomotion Walker2d | UTD Ratio | 1 |
| All Tasks | Batch Size | 256 |
| | Learning Rate | 3e-4 |
| | Weight Decay | 1e-3 |
| | Discount | 0.99 |
| | Hidden Dims | (256, 256) |
| | DAgger & HG-DAgger Pretrain Steps | 60,000 |
| | DAgger & HG-DAgger Steps Per Iteration | 2500 |

Table 5: RLIF and HG-DAgger parameters for each simulation task. The parameters specified under All Tasks are for both BC and RLIF.

| Tasks | Parameters | Values |
|---|---|---|
| Insertion and Unfolding Tasks on Franka Robot | Pretrained Vision Backbone | EfficientNet-B3 |
| | Learning Rate | 3e-4 |
| | MLP Dims | (256, 256) |
| | Layer Norm | true |
| | Discount | 0.99 |
| | UTD Ratio | 4 |
| | Batch Size | 256 |

Table 6: RLIF and HG-DAgger parameters for insertion task on Franka robot.

# B SUBOPTIMALITY GAP

## B.1 SUBOPTIMALITY GAP OF RLIF

**Theorem B.1** (Suboptimality Gap of RLIF, Thm. 6.3 restated). *Let $\tilde{\pi} \in \Pi_\delta^{\text{opt}}$ denote an optimal policy from maximizing the reward function $\tilde{r}_\delta$ generated by RLIF. Let $\epsilon = \max \left\{ \mathbb{E}_{s \sim d_\mu^{\tilde{\pi}}} \ell(s, \pi(s)), \mathbb{E}_{s \sim d_\mu^{\tilde{\pi}}} \ell'(s, \pi(s)) \right\}$ (Def. 6.2). Under Assumption 6.1, when $V^{\pi^{\text{ref}}}$ is known, the RLIF suboptimality gap satisfies:*

$$\text{SubOpt}_{\text{RLIF}} = V^\star(\mu) - V^{\tilde{\pi}}(\mu) \le \min \left\{ V^{\pi^\star}(\mu) - V^{\pi^{\text{ref}}}(\mu), V^{\pi^\star}(\mu) - V^{\pi^{\exp}}(\mu) \right\} + \frac{\delta\epsilon}{1-\gamma}.$$

*Proof.* Notice that $\tilde{\pi}$ denote the optimal policy w.r.t. the RLIF reward function $\tilde{r}_\delta$:

$$\tilde{\pi} \in \arg\max_{\pi \in \Pi} \mathbb{E}_{a_t \sim \pi(s_t), \tilde{r}_\delta(s_t, \pi(s_t))} \left[ \sum_{t=0}^\infty \gamma^t \tilde{r}_\delta(s_t, a_t) | s_0 \sim \mu \right]. \tag{B.1}$$

Let $\mathcal{E}$ denote the following event:

$$\mathcal{E} = \left\{ Q^{\pi^{\text{ref}}}(s, \pi^{\text{ref}}(s)) > Q^{\pi^{\text{ref}}}(s, \pi(s)) + \delta \text{ or } Q^{\pi^{\exp}}(s, \pi^{\exp}(s)) > Q^{\pi^{\exp}}(s, \pi(s)) + \delta \right\}.$$

By Assumption 6.1, we can write the reward functions as a random variable as follows:

$$\mathbb{P}(\tilde{r}_\delta(s, \pi(s)) = 0 | s) = \begin{cases} \beta, & \text{if } \mathcal{E} \text{ happens,} \\ 1 - \beta, & \text{otherwise.} \end{cases}$$

$$\mathbb{P}(\tilde{r}_\delta(s, \pi(s)) = 1 | s) = \begin{cases} \beta, & \text{if } \bar{\mathcal{E}} \text{ happens,} \\ 1 - \beta, & \text{otherwise.} \end{cases} \tag{B.2}$$

Taking an expectation of the equations above, we have

$$\mathbb{E}_{\tilde{r}_\delta(s, \pi(s))} \tilde{r}_\delta(s, \pi(s)) = 1 - \beta, \text{ when } \mathcal{E} \text{ happens,}$$
$$\mathbb{E}_{\tilde{r}_\delta(s, \pi(s))} \tilde{r}_\delta(s, \pi(s)) = \beta, \text{ when } \bar{\mathcal{E}} \text{ happens.} \tag{B.3}$$

And since we assume $\beta > 0.5$, we know that $\mathbb{E}_{\tilde{r}_\delta(s, \pi(s))} \tilde{r}_\delta(s, \pi(s))$ achieves maximum $\beta$, when $\bar{\mathcal{E}}$ happens. Hence, in order to maximize the overall return in Eqn. 6.1, $\tilde{\pi}$ should satisfy $Q^{\pi^{\text{ref}}}(s, \pi^{\text{ref}}(s)) \le Q^{\pi^{\text{ref}}}(s, \tilde{\pi}(s)) + \delta$ and $Q^{\pi^{\exp}}(s, \pi^{\exp}(s)) \le Q^{\pi^{\exp}}(s, \tilde{\pi}(s)) + \delta, \forall s \in \mathcal{S}$. Since this result holds for all states $s \in \mathcal{S}$, then for $\left| V^{\pi^{\text{ref}}}(\mu) - V^{\tilde{\pi}}(\mu) \right|$, we have

$$\left| V^{\pi^{\text{ref}}}(\mu) - V^{\tilde{\pi}}(\mu) \right| = \left| \mathbb{E}_{s \sim d_\mu^{\pi^{\text{ref}}}} Q^{\pi^{\text{ref}}}(s, \pi^{\text{ref}}(s)) - \mathbb{E}_{s \sim d_\mu^{\tilde{\pi}}} Q^{\tilde{\pi}}(s, \tilde{\pi}(s)) \right|$$

$$\le \left| \mathbb{E}_{s \sim d_\mu^{\pi^{\text{ref}}}} Q^{\pi^{\text{ref}}}(s, \pi^{\text{ref}}(s)) - \mathbb{E}_{s \sim d_\mu^{\tilde{\pi}}} Q^{\pi^{\text{ref}}}(s, \tilde{\pi}(s)) \right|$$

$$+ \left| \mathbb{E}_{s \sim d_\mu^{\tilde{\pi}}} Q^{\pi^{\text{ref}}}(s, \tilde{\pi}(s)) - \mathbb{E}_{s \sim d_\mu^{\tilde{\pi}}} Q^{\tilde{\pi}}(s, \tilde{\pi}(s)) \right|$$

$$\le \delta\epsilon + \gamma \left| \mathbb{E}_{s' \sim P(s, \tilde{\pi}(s))} \left[ \mathbb{E}_{s \sim d_\mu^{\pi^{\text{ref}}}} Q^{\pi^{\text{ref}}}(s', \pi^{\text{ref}}(s')) - \mathbb{E}_{s \sim d_\mu^{\tilde{\pi}}} Q^{\tilde{\pi}}(s', \tilde{\pi}(s')) \right] \right|$$

$$\cdots$$

$$\le \delta\epsilon + \gamma\delta\epsilon + \gamma^2\delta\epsilon + \cdots = \frac{\delta\epsilon}{1-\gamma}.$$

Applying a similar strategy to $\left| V^{\pi^{\exp}}(\mu) - V^{\tilde{\pi}}(\mu) \right|$, we can also get $\left| V^{\pi^{\exp}}(\mu) - V^{\tilde{\pi}}(\mu) \right| \le \frac{\delta\epsilon}{1-\gamma}$. Hence, we conclude the following

$$\max \left\{ \left| V^{\pi^{\text{ref}}}(\mu) - V^{\tilde{\pi}}(\mu) \right|, \left| V^{\pi^{\exp}}(\mu) - V^{\tilde{\pi}}(\mu) \right| \right\} \le \frac{\delta\epsilon}{1-\gamma}. \tag{B.4}$$

Hence, we conclude that

$$V^\star(\mu) - V^{\tilde{\pi}}(\mu)$$

$$\le \min \left\{ V^{\pi^\star}(\mu) - V^{\pi^{\exp}}(\mu) + \left| V^{\pi^{\exp}}(\mu) - V^{\tilde{\pi}}(\mu) \right|, V^{\pi^\star}(\mu) - V^{\pi^{\text{ref}}}(\mu) + \left| V^{\pi^{\text{ref}}}(\mu) - V^{\tilde{\pi}}(\mu) \right| \right\}$$

$$\le \min \left\{ V^{\pi^\star}(\mu) - V^{\pi^{\text{ref}}}(\mu), V^{\pi^\star}(\mu) - V^{\pi^{\exp}}(\mu) \right\} + \frac{\delta\epsilon}{1-\gamma},$$

where the last inequality holds due to Eqn. B.4. Hence we conclude the results.

$\square$

## B.2 SUBOPTIMALITY GAP OF DAGGER (ROSS ET AL., 2011)

**Corollary B.2** (Suboptimality Gap of DAgger, Cor. 6.4 restated). *Let $\tilde{\pi} \in \Pi_\delta^{\mathrm{opt}}$ denote an optimal policy from maximizing the reward function $\tilde{r}_\delta$ generated by RLIF. Let $\epsilon = \mathbb{E}_{s \sim d_\mu^{\tilde{\pi}}} \ell(s, \pi(s))$ (Def. 6.2). Under Assumption 6.1, when $V^{\pi^{\mathrm{ref}}}$ is unknown, the DAgger suboptimality gap satisfies:*

$$\mathrm{SubOpt}_{\mathrm{DAgger}} = V^\star(\mu) - V^{\tilde{\pi}}(\mu) \leq V^{\pi^\star}(\mu) - V^{\pi^{\mathrm{exp}}}(\mu) + \frac{\delta \epsilon}{1 - \gamma}. \tag{B.5}$$

*Proof.* Notice that $\tilde{\pi}$ denote the optimal policy w.r.t. the RLIF reward function $\tilde{r}_\delta$:

$$\tilde{\pi} \in \arg\max_{\pi \in \Pi} \mathbb{E}_{a_t \sim \pi(s_t), \tilde{r}_\delta(s_t, \pi(s_t))} \left[ \sum_{t=0}^\infty \gamma^t \tilde{r}_\delta(s_t, a_t) | s_0 \sim \mu \right]. \tag{B.6}$$

Let $\mathcal{E}$ denote the following event:

$$\mathcal{E} = \left\{ Q^{\pi^{\mathrm{exp}}}(s, \pi^{\mathrm{exp}}(s)) > Q^{\pi^{\mathrm{exp}}}(s, \pi(s)) + \delta \right\}.$$

By Assumption 6.1, we can write the reward functions as a random variable as follows:

$$\mathbb{P}(\tilde{r}_\delta(s, \pi(s)) = 0 | s) = \begin{cases} \beta, & \text{if } \mathcal{E} \text{ happens}, \\ 1 - \beta, & \text{otherwise}. \end{cases}$$

$$\mathbb{P}(\tilde{r}_\delta(s, \pi(s)) = 1 | s) = \begin{cases} \beta, & \text{if } \bar{\mathcal{E}} \text{ happens}, \\ 1 - \beta, & \text{otherwise}. \end{cases} \tag{B.7}$$

Taking an expectation of the equations above, we have

$$\mathbb{E}_{\tilde{r}_\delta(s, \pi(s))} \tilde{r}_\delta(s, \pi(s)) = 1 - \beta, \text{ when } \mathcal{E} \text{ happens},$$
$$\mathbb{E}_{\tilde{r}_\delta(s, \pi(s))} \tilde{r}_\delta(s, \pi(s)) = \beta, \text{ when } \bar{\mathcal{E}} \text{ happens}. \tag{B.8}$$

And since we assume $\beta > 0.5$, we know that $\mathbb{E}_{\tilde{r}_\delta(s, \pi(s))} \tilde{r}_\delta(s, \pi(s))$ achieves maximum $\beta$, when $\bar{\mathcal{E}}$ happens. Hence, in order to maximize the overall return in Eqn. 6.1, $\tilde{\pi}$ should satisfy $Q^{\pi^{\mathrm{exp}}}(s, \pi^{\mathrm{exp}}(s)) \leq Q^{\pi^{\mathrm{exp}}}(s, \tilde{\pi}(s)) + \delta, \forall s \in \mathcal{S}$. Since this result holds for all states $s \in \mathcal{S}$, then for $\left| V^{\pi^{\mathrm{exp}}}(\mu) - V^{\tilde{\pi}}(\mu) \right|$, we have

$$\left| V^{\pi^{\mathrm{exp}}}(\mu) - V^{\tilde{\pi}}(\mu) \right| = \left| \mathbb{E}_{s \sim d_\mu^{\pi^{\mathrm{exp}}}} Q^{\pi^{\mathrm{exp}}}(s, \pi^{\mathrm{exp}}(s)) - \mathbb{E}_{s \sim d_\mu^{\tilde{\pi}}} Q^{\tilde{\pi}}(s, \tilde{\pi}(s)) \right|$$

$$\leq \left| \mathbb{E}_{s \sim d_\mu^{\pi^{\mathrm{exp}}}} Q^{\pi^{\mathrm{exp}}}(s, \pi^{\mathrm{exp}}(s)) - \mathbb{E}_{s \sim d_\mu^{\tilde{\pi}}} Q^{\pi^{\mathrm{exp}}}(s, \tilde{\pi}(s)) \right|$$

$$+ \left| \mathbb{E}_{s \sim d_\mu^{\tilde{\pi}}} Q^{\pi^{\mathrm{exp}}}(s, \tilde{\pi}(s)) - \mathbb{E}_{s \sim d_\mu^{\tilde{\pi}}} Q^{\tilde{\pi}}(s, \tilde{\pi}(s)) \right|$$

$$\leq \delta \epsilon + \gamma \left| \mathbb{E}_{s' \sim P(s, \tilde{\pi}(s))} \left[ \mathbb{E}_{s \sim d_\mu^{\pi^{\mathrm{exp}}}} Q^{\pi^{\mathrm{exp}}}(s', \pi^{\mathrm{exp}}(s')) - \mathbb{E}_{s \sim d_\mu^{\tilde{\pi}}} Q^{\tilde{\pi}}(s', \tilde{\pi}(s')) \right] \right| \tag{B.9}$$

$$\cdots$$

$$\leq \delta \epsilon + \gamma \delta \epsilon + \gamma^2 \delta \epsilon + \cdots = \frac{\delta \epsilon}{1 - \gamma}.$$

Hence, we conclude the following

$$V^\star(\mu) - V^{\tilde{\pi}}(\mu)$$

$$\leq V^{\pi^\star}(\mu) - V^{\pi^{\mathrm{exp}}}(\mu) + \left| V^{\pi^{\mathrm{exp}}}(\mu) - V^{\tilde{\pi}}(\mu) \right|$$

$$\leq V^{\pi^\star}(\mu) - V^{\pi^{\mathrm{exp}}}(\mu) + \frac{\delta \epsilon}{1 - \gamma},$$

where the last inequality holds due to Eqn. B.9.

$\square$

## B.3 TIGHT SUBOPTIMALITY GAP

**Example B.3** (Lower Bounds of RLIF). *Consider a bandit problem where we only have one state $\mathcal{S} = \{s\}$, two actions $\mathcal{A} = \{a_1, a_2\}$ and, and the reward function is given as $r(s, a_1) = 1$ and $r(s, a_2) = 0$. Assume the policy space $\Pi \subset \mathbb{R}_+^2$ is a two-dimensional Euclidean space, where each policy $\pi \in \Pi$ satisfies such that $\pi(a_1) + \pi(a_2) = 1$. Let $\pi^{\text{ref}}, \pi^{\text{exp}} \in \Pi$ be any two policies in the policy space, and $\tilde{\pi} \in \Pi_\delta^{\text{opt}}$ be the optimal policy from maximizing the reward function $\tilde{r}_\delta$ generated by RLIF, $\epsilon = \max\left\{\mathbb{E}_{s \sim d_\mu^{\tilde{\pi}}} \ell(s, \pi(s)), \mathbb{E}_{s \sim d_\mu^{\tilde{\pi}}} \ell'(s, \pi(s))\right\}$ (Def. 6.2). Then we have*

$$\text{SubOpt}_{\text{RLIF}} = \min\left\{V^{\pi^\star}(\mu) - V^{\pi^{\text{ref}}}(\mu), V^{\pi^\star}(\mu) - V^{\pi^{\text{exp}}}(\mu)\right\} + \frac{\delta\epsilon}{1 - \gamma},$$

$$\text{SubOpt}_{\text{DAgger}} = V^{\pi^\star}(\mu) - V^{\pi^{\text{exp}}}(\mu) + \frac{\delta\epsilon}{1 - \gamma}.$$

*Proof.* Note that in the bandit case constructed above, the optimal policy should satisfy $\pi^\star = [1, 0]^\top$. Let $\pi^{\text{ref}}, \pi^{\text{exp}}$ be that

$$\pi^{\text{ref}} = [x_1, 1 - x_1]^\top, \ \pi^{\text{exp}} = [x_2, 1 - x_2]^\top. \tag{B.10}$$

Hence, we know that

$$V^{\pi^{\text{ref}}}(s) = \frac{x_1}{1 - \gamma}, \ V^{\pi^{\text{ref}}}(s) = \frac{x_2}{1 - \gamma}. \tag{B.11}$$

Let $\mathcal{E}$ denote the following event:

$$\mathcal{E} = \left\{Q^{\pi^{\text{ref}}}(s, \pi^{\text{ref}}(s)) > Q^{\pi^{\text{ref}}}(s, \pi(s)) + \delta \text{ or } Q^{\pi^{\text{exp}}}(s, \pi^{\text{exp}}(s)) > Q^{\pi^{\text{exp}}}(s, \pi(s)) + \delta\right\}.$$

By Assumption 6.1, we can write the reward functions as a random variable as follows:

$$\mathbb{P}(\tilde{r}_\delta(s, \pi(s)) = 0 | s) = \begin{cases} \beta, & \text{if } \mathcal{E} \text{ happens,} \\ 1 - \beta, & \text{otherwise.} \end{cases}$$
$$\mathbb{P}(\tilde{r}_\delta(s, \pi(s)) = 1 | s) = \begin{cases} \beta, & \text{if } \bar{\mathcal{E}} \text{ happens,} \\ 1 - \beta, & \text{otherwise.} \end{cases} \tag{B.12}$$

Taking an expectation of the equations above, we have

$$\mathbb{E}_{\tilde{r}_\delta(s, \pi(s))} \tilde{r}_\delta(s, \pi(s)) = 1 - \beta, \text{ when } \mathcal{E} \text{ happens,}$$
$$\mathbb{E}_{\tilde{r}_\delta(s, \pi(s))} \tilde{r}_\delta(s, \pi(s)) = \beta, \text{ when } \bar{\mathcal{E}} \text{ happens.} \tag{B.13}$$

And since we assume $\beta > 0.5$, we know that $\mathbb{E}_{\tilde{r}_\delta(s, \pi(s))} \tilde{r}_\delta(s, \pi(s))$ achieves maximum $\beta$, when $\bar{\mathcal{E}}$ happens. Hence, in order to maximize the overall return in Eqn. 6.1, $\tilde{\pi} = [x_3, 1 - x_3]^\top$ should satisfy $Q^{\pi^{\text{ref}}}(s, \pi^{\text{ref}}(s)) \leq Q^{\pi^{\text{ref}}}(s, \tilde{\pi}(s)) + \delta$ and $Q^{\pi^{\text{exp}}}(s, \pi^{\text{exp}}(s)) \leq Q^{\pi^{\text{exp}}}(s, \tilde{\pi}(s)) + \delta, \forall s \in \mathcal{S}$, which implies that

$$x_3 + \delta \geq x_1, \ x_3 + \delta \geq x_2. \tag{B.14}$$

Without loss of generality, assume $x_1 \geq x_2$. Let $x_3 = x_1 + \delta$, then we have

$$\text{SubOpt}_{\text{RLIF}} = \mathbb{E}\left\{V^\star(s) - V^{\tilde{\pi}}(s)\right\} = \mathbb{E}_{s \sim d_\mu^{\tilde{\pi}}}\left[\frac{1 - x_3}{1 - \gamma}\right] = \frac{1 - x_1}{1 - \gamma} + \frac{\delta\epsilon}{1 - \gamma} \tag{B.15}$$

$$= \min\left\{V^{\pi^\star}(s) - V^{\pi^{\text{ref}}}(s), V^{\pi^\star}(s) - V^{\pi^{\text{exp}}}(s)\right\} + \frac{\delta\epsilon}{1 - \gamma}. \tag{B.16}$$

In the DAgger case, by setting $\pi^{\text{ref}} = \pi^{\text{exp}}$, we can similarly obtain

$$\text{SubOpt}_{\text{RLIF}} = \mathbb{E}\left\{V^\star(s) - V^{\tilde{\pi}}(s)\right\} = \mathbb{E}_{s \sim d_\mu^{\tilde{\pi}}}\left[\frac{1 - x_3}{1 - \gamma}\right] = \frac{1 - x_1}{1 - \gamma} + \frac{\delta\epsilon}{1 - \gamma} \tag{B.17}$$

$$= V^{\pi^\star}(s) - V^{\pi^{\text{exp}}}(s) + \frac{\delta\epsilon}{1 - \gamma}, \tag{B.18}$$

Hence, we conclude our results. $\square$

## C  NON-ASYMPTOTIC ANALYSIS

For the non-asymptotic analysis, we adopt the LCB-VI (Rashidinejad et al., 2021; Xie et al., 2021; Li et al., 2022a) framework, because we warm-start the training process by adding a small amount of offline data, and later on we gradually mix $d_\mu^{\pi^{\exp}}$ into the replay buffer due to the intervention. The LCB-VI framework provides a useful tool for studying the distribution shift in terms of the concentrability coefficient when incorporating offline data. While RLPD (Ball et al., 2023) is not an LCB-VI algorithm, our method is generic with respect to the choice of RL subroutines, such as the online setting (Azar et al., 2017; Gupta et al., 2022) or the hybrid setting (Song et al., 2022). Hence we believe this analysis is useful for characterizing our approach.

We first present the single-policy concentrability coefficient that has been widely studied in prior literature (Rashidinejad et al., 2021; Li et al., 2022a) as the following.

**Definition C.1** (Single-Policy Concentrability Coefficient). *We define the concentrability coefficient of $d^{\pi^\star}$ with respect to an offline visitation distribution $\rho$ is defined as $C_\rho^\star := \max_{(s,a) \in \mathcal{S} \times \mathcal{A}} \frac{d_\mu^{\pi^\star}(s,a)}{\rho(s,a)}$. Similar to Li et al. (2022a), we adopt the conventional setting of $d_\mu^{\pi^\star}(s,a)/\rho(s,a) = 0$, when $d_\mu^{\pi^\star}(s,a) = 0$ for a state-action pair $(s,a)$.*

Note that the concentrability coefficient plays a crucial role in the final non-asymptotic bound, we present an upper bound on the concentrability coefficient with RLIF.

**Lemma C.2** (Concentrablity of a Intervention Probability). *Suppose we construct $\mu^{\mathrm{int}}$ by mixing $d^{\pi^{\mathrm{ref}}}$ with probability $\beta$ $(0 < \beta < 1)$ and distribution $\rho$ such that $\rho(s,a) > 0, \forall (s,a) \in \mathcal{S} \times \mathcal{A}$: $\mu^{\mathrm{int}} = (1-\beta)\rho + \beta d^{\pi^{\mathrm{ref}}}$, then the concentrability coefficient satisfies $C_{\mu^{\mathrm{int}}}^\star \leq \min\left\{\frac{1}{1-\beta}C_\rho^*, \frac{1}{\beta}C_{\exp}^*\right\}$.*

*Proof.* Notice that when the optimal policy $\pi^\star$ is deterministic (Bertsekas, 2019; Li et al., 2022a), we can apply a similar decomposition technique shown in Li et al. (2023), we have

$$
\begin{aligned}
\max_{s,a} \frac{d_\mu^{\pi^\star}(s,a)}{\mu^{\mathrm{int}}(s,a)} &= \max_s \frac{d_\mu^{\pi^\star}(s)\mathbf{1}\left\{a = \pi^\star(s)\right\}}{\mu^{\mathrm{int}}(s)} \\
&= \left\|\frac{d_\mu^{\pi^\star}}{\mu^{\mathrm{int}}}\right\|_\infty \leq \left\|\frac{d^{\pi^\star}}{(1-\beta)\rho + \beta d_\mu^{\pi^{\mathrm{ref}}}}\right\|_\infty \leq \min\left\{\frac{1}{1-\beta}C_\rho^*, \frac{1}{\beta}C_{\exp}^*\right\}.
\end{aligned}
\tag{C.1}
$$

Hence, we conclude that

$$
C_{\mu^{\mathrm{int}}}^\star = \max_{s,a} \frac{d^{\pi^\star}(s,a)}{\mu^{\mathrm{int}}(s,a)} \leq \min\left\{\frac{1}{1-\beta}C_\rho^*, \frac{1}{\beta}C_{\exp}^*\right\}.
\tag{C.2}
$$

$\square$

With Lemma C.2, we present our main non-asymptotic sample complexity bound in the following.

**Corollary C.3** (Non-Asymptotic Sample Complexity). *Suppose the conditions in Assumption 6.1 holds, then there exists an algorithm that returns an $\epsilon$-optimal $\hat{\pi}$ for $V_{\tilde{r}_\delta}^\pi(\mu)$ such that $V_{\tilde{r}_\delta}^{\tilde{\pi}}(\mu) - V_{\tilde{r}_\delta}^{\hat{\pi}}(\mu) \leq \epsilon$, with a sample complexity of $\widetilde{O}\left(\frac{SC_{\exp}^\star}{(1-\gamma)^3\epsilon^2}\right)$.*

*Proof.* For any policy $\pi$, since RLIF will induce a dataset with the distribution of

$$
\mu^{\mathrm{int}} = (1-\beta)d_\mu^\pi + \beta d_\mu^{\pi^{\mathrm{ref}}}.
\tag{C.3}
$$

We overload the notation let $C_\pi^\star = C_{d_\mu^\pi}^\star$ to denote the concentrability coefficient w.r.t. the state visitation distribution $d_\mu^\pi$. Now applying Lemma C.2 with $\mu^{\mathrm{int}}$ defined in Eqn. C.3, we obtain a concentrability coefficient of

$$
C_{\mu^{\mathrm{int}}}^\star \leq \min\left\{\frac{1}{1-\beta}C_\pi^\star, \frac{1}{\beta}C_{\exp}^\star\right\} \leq \frac{1}{\beta}C_{\exp}^\star
\tag{C.4}
$$

Before applying such a concentrability coefficient to Thm. D.2, we will first provide an error bound induced by the stochastic reward $\tilde{r}_\delta$. $\forall \epsilon > 0$, suppose we want to achieve a statistical error of $\epsilon/2$ induced by the stochastic $\tilde{r}_\delta \in [0, 1]$. $\forall \eta > 0$ Proposition D.1 implies that

$$\mathbb{P}\left[\frac{1}{N}\left|\sum_{i=1}^{N}(\tilde{r}_\delta(s, \pi(s))_i - \mathbb{E}\tilde{r}_\delta(s, \pi(s)))\right| \geq \eta\right] \leq 2\exp\left[-2\eta^2 N\right], \ \forall s \in \mathcal{S}, \qquad \text{(C.5)}$$

where $\tilde{r}_\delta(s, \pi(s))_i$ is the $i^{th}$ sample of the reward function $\tilde{r}_\delta(s, \pi(s))$. Taking a union bound of the probability above overall $s \in \mathcal{S}$ yields

$$\mathbb{P}\left[\max_{s \in \mathcal{S}} \frac{1}{N}\left|\sum_{i=1}^{N}(\tilde{r}_\delta(s, \pi(s))_i - \mathbb{E}\tilde{r}_\delta(s, \pi(s)))\right| \geq \eta\right] \leq 2S\exp\left(-2\eta^2 N\right). \qquad \text{(C.6)}$$

Hence, by setting $\eta = (1-\gamma)\epsilon/2$, we know that with probability at least $1 - \delta_0$, the sub-optimality gap induced by the stochastic reward is at most $\epsilon/2$, given enough sample size $N = O\left(\frac{1}{(1-\gamma)^2\epsilon^2}\log\left(\frac{2S}{\delta_0}\right)\right) = \widetilde{O}\left(\frac{1}{(1-\gamma)^2\epsilon^2}\right)$ for one state $s \in \mathcal{S}$, which leads to the overall sample complexity of $\widetilde{O}\left(\frac{S}{(1-\gamma)^2\epsilon^2}\right)$. In the deterministic reward case, Thm. D.2 implies that one can find an algorithm that achieves a suboptimality gap of $\epsilon/2$ with the sample complexity of

$$\widetilde{O}\left(\frac{SC^\star_{\mu^{\text{int}}}}{(1-\gamma)^3\epsilon^2}\right) = \widetilde{O}\left(\frac{SC^\star_{\exp}}{\beta(1-\gamma)^3\epsilon^2}\right) = \widetilde{O}\left(\frac{SC^\star_{\exp}}{(1-\gamma)^3\epsilon^2}\right), \qquad \text{(C.7)}$$

where the above equation holds due to Eqn. C.4. Hence, we conclude there exists an algorithm that can achieve an $\epsilon$ optimal policy w.r.t. $V^\pi_{\tilde{r}_\delta}(\mu)$ at the total sample complexity of

$$\widetilde{O}\left(\frac{SC^\star_{\exp}}{(1-\gamma)^3\epsilon^2}\right) + \widetilde{O}\left(\frac{S}{(1-\gamma)^2\epsilon^2}\right) = \widetilde{O}\left(\frac{SC^\star_{\exp}}{(1-\gamma)^3\epsilon^2}\right). \qquad \text{(C.8)}$$

$\square$

Cor. C.3 indicates that the total sample complexity for learning $\hat{\pi}$ that maximizes $V^\pi_{\tilde{r}_\delta}$, the total sample complexity *does not exceed* the sample complexity $\widetilde{O}\left(\frac{SC^\star_{\exp}}{(1-\gamma)^3\epsilon^2}\right)$ of solving original RL problem with a true reward $r$.

# D    SUPPORTING THEORETICAL RESULTS

**Proposition D.1** (Hoeffding Bound, Proposition 2.5 of Wainwright (2019)). *Suppose that variables* $x_i, i = 1, 2, \ldots, n$ *are independent, and* $x_i \in [a, b], \forall i = 1, 2, \ldots, n$. *Then for all* $t > 0$, *we have*

$$\mathbb{P}\left[\left|\sum_{i=1}^{n}\frac{1}{n}(x_i - \mathbb{E}x_i)\right| \geq t\right] \leq 2\exp\left[-\frac{nt^2}{(b-a)^2}\right]. \qquad \text{(D.1)}$$

**Theorem D.2** (Convergence of Offline RL, Thm. 1 of Li et al. (2022a)). *Suppose* $\gamma \in [1/2, 1)$, $\epsilon \in \left(0, \frac{1}{1-\gamma}\right]$ *and the concentrability coefficient* $C^\star_\rho$ *is defined in Def. C.1. With high probability, there exists an algorithm that learns an algorithm* $\hat{\pi}$ *with concentrability coefficient* $C^\star_\rho$, *that* $\hat{\pi}$ *can achieve an $\epsilon$-optimal policy with a sample complexity of* $N = \widetilde{O}\left(\frac{SC^\star_\rho}{(1-\gamma)^3\epsilon^2}\right)$, *in terms of the tuples* $\{s_i, a_i, s'_i\}_{i=1}^{N}$, *where* $(s_i, a_i) \sim \rho, \forall(s, a) \in \mathcal{S} \times \mathcal{A}$.

# E    PROPERTIES OF $\Pi^{\text{opt}}_\delta$.

**Lemma E.1** (Properties of $\Pi^{\text{opt}}_\delta$). $\forall \delta > 0$, $d_\delta$ *is a metric over* $\Pi^{\text{opt}}_\delta$. *Let*

$$\left|\Pi^{\text{opt}}_\delta\right| := \max_{\pi, \pi' \in \Pi^{\text{opt}}_\delta} d_\delta(\pi, \pi'), \qquad \text{(E.1)}$$

*then* $\left|\Pi^{\text{opt}}_\delta\right|$ *is monotonic decreasing in* $\delta$.

*Proof.* To verify that $\Pi_\delta^{\mathrm{opt}}$ is a metric, we need to show the following properties:

- Distance to to itself is zero: $\forall \pi \in \Pi_\delta^{\mathrm{opt}}$, we have that

$$d_\delta(\pi, \pi') = \max_{s \in \mathcal{S}} \|\pi(\cdot|s) - \pi(\cdot|s)\|_1 = 0. \tag{E.2}$$

- Positivity: $\forall \pi, \pi' \in \Pi_\delta^{\mathrm{opt}}$ such that $\pi \neq \pi'$, we have that

$$d_\delta(\pi, \pi') = \max_{s \in \mathcal{S}} \|\pi(\cdot|s) - \pi'(\cdot|s)\|_1 > 0. \tag{E.3}$$

- Symmetry: $\forall \pi, \pi' \in \Pi_\delta^{\mathrm{opt}}$, we have that

$$d_\delta(\pi, \pi') = \max_{s \in \mathcal{S}} \|\pi(\cdot|s) - \pi'(\cdot|s)\|_1 = \|\pi'(\cdot|s) - \pi(\cdot|s)\|_1 = d_\delta(\pi', \pi). \tag{E.4}$$

- Triangle inequality: $\forall \pi_1, \pi_2, \pi_3 \in \Pi_\delta^{\mathrm{opt}}$, we have

$$
\begin{aligned}
d_\delta(\pi_1, \pi_3) &= \max_{s \in \mathcal{S}} \|\pi_1(\cdot|s) - \pi_3(\cdot|s)\|_1 \\
&\leq \max_{s \in \mathcal{S}} \left[ \|\pi_1(\cdot|s) - \pi_2(\cdot|s)\|_1 + \|\pi_2(\cdot|s) - \pi_3(\cdot|s)\|_1 \right] \\
&\leq \max_{s \in \mathcal{S}} \|\pi_1(\cdot|s) - \pi_2(\cdot|s)\|_1 + \max_{s \in \mathcal{S}} \|\pi_2(\cdot|s) - \pi_3(\cdot|s)\|_1 \\
&= d_\delta(\pi_1, \pi_2) + d_\delta(\pi_2, \pi_3).
\end{aligned} \tag{E.5}
$$

Hence we conclude that $d_\delta$ is a metric over $\Pi_\delta^{\mathrm{opt}}$. Next, we will show $\left|\Pi_\delta^{\mathrm{opt}}\right|$ is monotonic decreasing. We will show this result by proving $\Pi_\delta^{\mathrm{opt}} \subset \Pi_{\delta'}^{\mathrm{opt}}$, $\forall 0 < \delta < \delta'$. Notice that $\forall \pi \in \Pi_\delta^{\mathrm{opt}}$, we have that

$$
\begin{aligned}
V_{\tilde{r}_\delta}^\pi(\mu) &= \max_\pi \mathbb{E}_{a_t \sim \pi(s_t), \tilde{r}_\delta(s_t, \pi(s_t))} \left[ \sum_{t=0}^\infty \gamma^t \tilde{r}_\delta(s_t, a_t) | s_0 \sim \mu \right] \\
&= \frac{1}{1 - \gamma} = \max_\pi \mathbb{E}_{a_t \sim \pi(s_t), \tilde{r}_{\delta'}(s_t, \pi(s_t))} \left[ \sum_{t=0}^\infty \gamma^t \tilde{r}_{\delta'}(s_t, a_t) | s_0 \sim \mu \right].
\end{aligned} \tag{E.6}
$$

This results implies that $\pi \in \Pi_{\delta'}^{\mathrm{opt}}$. Therefore, we know that $\left|\Pi_\delta^{\mathrm{opt}}\right| \leq \left|\Pi_{\delta'}^{\mathrm{opt}}\right|$, when $\delta \leq \delta'$. Hence, we conclude the our results. $\qquad \square$

