# OpenReview forum: "RLIF: Interactive Imitation Learning as Reinforcement Learning"
_ICLR.cc/2024/Conference — ICLR 2024 poster_

### Official Review · Reviewer_48gS · 2023-10-18

**Soundness:** 3 good
**Presentation:** 4 excellent
**Contribution:** 3 good
**Rating:** 8
**Confidence:** 4

**Summary:**

This paper provides a methodology for learning a policy given human intervention. Unlike other imitation learning frameworks such as Dagger, the method does not directly clone the actions taken by the expert, but rather uses the expert’s decision to intervene as a negative reward signal. The paper claims that by doing so, the learned policy has the potential to exceed the performance of a suboptimal expert, and does not need to assume the expert is optimal. The method is demonstrated experimentally in simulation to outperform imitation learning baselines, and is demonstrated to work in a real world experiment. Theoretical analysis is provided to justify the method.

**Strengths:**

The key strengths of the paper are as follows:
- The method is simple and easy to implement, thereby making it useful.
- The method does not require access to a task reward signal and makes only the benign assumption that an expert would intervene if the policy were performing badly. This is realistic and gives the method a broad scope.
- The paper includes a thorough justification of the method, which with some additional clarifications could be compelling (see the questions I have).

**Weaknesses:**

The main weakness of this paper is that I am suspicious of the results due to some confusion. Please answer the questions below, and I would be happy to revise my rating upward if they are satisfactory. [Edit: updated rating since questions were addressed]

Aside from the questions, below are a few recommendations for clarity improvement and a few grammatical errors caught.

Clarity recommendations:
- In the intro, add a more intuitive explanation for how RLIF is able to exceed the expert’s performance without access to a task reward signal.
- In the intro, you describe the theoretical analysis performed, but don't state the bottom line. What does the analysis tell us?

Grammar:
- “…for selecting when to intervene lead to good performance…”
- “We leave the of DAgger analysis under…”

**Questions:**

- If the expert intervenes for 5 steps, would all 5 transitions be labeled -1? Wouldn’t this result in actions taken by the expert being labeled -1 too?
- Could the else statement in line 7 of Algorithm 2 set the reward to the RL reward signal if it were available?
- I’m very confused by this sentence: “the performance of DAgger-like algorithms will be subject to the suboptimality of the experts, while our method can reach good performance even with suboptimal experts by learning from the expert’s decision of when to intervene.“ If no task reward is provided, how could the policy end up outperforming the expert?
- How do you end up with a 110% expert if the expert level is w.r.t an expert trained to near optimality?
- It seems unfair that Dagger variants are given 100 episodes to learn from while RLIF gets 1 million. It is also unrealistic to expect that we could have 1 million episodes in any real-world setting. How would RLIF perform if also restricted to a more realistic 10-100 episodes?
- In experiments, do you warm start the dataset for RLIF? If so, with what reward? And does this pose an unfair advantage to your algorithm?

---

> ### Author Response · Authors · 2023-11-19
> **Response 1/2**
>
> Dear Reviewer 48gS,
>
> We would like to express our gratitude for your insightful feedback and acknowledgment of our work. We have carefully considered your questions and provided clarifications below, which will be incorporated into the paper to enhance clarity. We believe that these modifications and clarifications address the weakness you point out about confusion in the paper. Please let us know if all issues have been addressed, or if any further issues remain, as we would be happy to fix these!
>
> > If the expert intervenes for 5 steps, would all 5 transitions be labeled -1? Wouldn’t this result in actions taken by the expert being labeled -1 too?
>
> Expert actions won’t be labeled as -1 when an expert intervenes for 5 steps, only the state and action pair immediately before the intervention begins is labeled -1, this was originally in algorihtm 2 box. The intuition here is that the -1 reward is assigned to the transition (state-action tuple) that most immediately resulted in the intervention. Our theoretical analysis in Section 6 shows that this leads to a good policy (that approximately maximizes the true reward) in theory, and our empirical evaluation shows this in practice. Expert actions are labeled as 0, similar to non-intervention transitions.
>
> > Could the else statement in line 7 of Algorithm 2 set the reward to the RL reward signal if it were available?
>
> Yes, that could be a straightforward extension as you suggested. Though in our paper we focused on learning entirely from intervention feedback without knowledge of the ground truth reward.  It’s reasonable to assume the sample efficiency of RLIF would be further improved if it could also access the true task reward.
>
> > I’m very confused by this sentence: “the performance of DAgger-like algorithms will be subject to the suboptimality of the experts, while our method can reach good performance even with suboptimal experts by learning from the expert’s decision of when to intervene.“ If no task reward is provided, how could the policy end up outperforming the expert?
>
> A crucial assumption for RLIF to work is that the intervention signal itself should contain task-relevant information. For example, in the driving example in the introduction, it is easier for a safety driver to intervene when the car does something bad than it is for them to carry out a perfectly optimal sequence of intervention actions – the choice of when to intervene itself often carries a lot of information. In Section 5.1, we provide one example of an expert intervention model based on Q-values that would carry considerable information in this choice. we also ablate the performance of RLIF  w.r.t. Q functions containing different level of task information; as you hypothesized, the least task-relevant information Q function has, the worse the performance, as in Appendix A.6.  Of course, there is no free lunch, and our method is sensitive to the particular intervention strategy the human uses, as evidenced by the fact that the naive random intervention baseline in our experiments performs much worse. However, this also indicates that our method is effective at deriving a learning signal from the intervention strategy, and the real-world experiments illustrate that real humans do provide interventions that are informative.
>
> > How do you end up with a 110% expert if the expert level is w.r.t an expert trained to near optimality?
>
> The statement about a 110% expert level refers to the expert's evaluation normalized score. In D4RL locomotion environments, we additionally learn expert policies from a curated dataset, so it can be possible that learned expert achieve normalized scores above 100.  And we label them as greater than 100% in expert levels. This was also explained Section 5.2 - results and discussions. We’ll add a more detailed explanation in the final paper.

---

> > ### Author Response · Authors · 2023-11-19
> > **Response 2/2**
> >
> > > It seems unfair that Dagger variants are given 100 episodes to learn from while RLIF gets 1 million. It is also unrealistic to expect that we could have 1 million episodes in any real-world setting. How would RLIF perform if also restricted to a more realistic 10-100 episodes?
> >
> > Thank you for the observation, this is indeed an oversight on our side!
> > We would like to clarify that, for RLIF it’s 1M environment steps, so it is 5000 episodes.
> > We chose 500 episodes for DAgger because the performance of DAgger converged at this point and stopped changing. To show this, we also present results for the Adroit Pen environment for 5000 episodes below, it’s indeed not doing better than training for 500 episodes.
> > Additionally, RLIF converges around 500 episodes or earlier most of the time as seen in Appendix A.5.
> >
> > |Adroit Pen          | HG-DAgger 5000 episodes| HG-DAgger 500 episodes|
> > |--------------------|---------------------------|---------------------------|
> > | 90% Expert         | 76.47                     | 73.47    |
> > | 40% Expert         | 51.93                     | 60   |
> > | 8% Expert          | 28.07                     | 28.53  |
> >
> >
> > Additionally, in our real-world robot experiments in Figure 5, DAgger and RLIF were both trained for fewer than 10 iterations, which amounts to less than an hour of wall clock time. We found RLIF performs better than DAgger in these complex vision-based real-world robotic tasks.
> >
> >
> > > In experiments, do you warm start the dataset for RLIF? If so, with what reward? And does this pose an unfair advantage to your algorithm?
> >
> > RLIF's replay buffer is warm-started with expert demonstrations labeled with zero reward, ensuring a fair comparison with DAgger-based methods, which also use the same offline dataset.

---

> > > ### Comment · Reviewer_48gS · 2023-11-20
> > > **Thanks for your comments**
> > >
> > > Thanks for your clarifications. I've updated my recommendation.

---

> > > > ### Author Response · Authors · 2023-11-20
> > > >
> > > > Thank you!

---

### Official Review · Reviewer_guRK · 2023-10-29

**Soundness:** 3 good
**Presentation:** 3 good
**Contribution:** 3 good
**Rating:** 6
**Confidence:** 3

**Summary:**

The paper tackles the problem of learning from suboptimal experts in interactive settings. It proposes to add a negative reward term in reinforcement learning whenever an expert requires being invoked in a RL DAgger setup, and it shows performance gains with different expert accuracy levels compared to supervised learning baselines.

**Strengths:**

- I appreciate the effort to bring the model to a real-world setting.

Edit:
- The paper has a through theoretical justification and a sensible set of simulated baselines, showing the power of the proposed method.

**Weaknesses:**

- The paper only compares with DAgger in supervised learning setups, even though DAgger is often used in RL and should have been included as a baseline in such configuration.
- The proposed technique seems to reduce to a heuristic to define a reward function.

Edit:
- After a very thorough set of responses from the authors, I admit I misinterpreted the actual scope of the paper, so these weaknesses are actually obsolete.

**Questions:**

- What is the advantage of the proposed paper versus using DAgger in RL?

---

> ### Author Response · Authors · 2023-11-19
>
> Dear reviewer guRK,
>
> Thank you for your thoughtful feedback, we would like to address your concerns below.
>
> >DAgger in RL setting
>
> We compare to DAgger under the same assumptions as our method -- namely, for both DAgger and RLIF, no ground truth reward function is provided, and the algorithm performs rollouts from the current policy with interventions provided by an expert. Could you elaborate on what kind of comparison you would like to see here? We are not aware of any prior work that uses DAgger in RL, but it's entirely possible we missed something (or misunderstood your comment), so if you can provide a reference to a relevant method we should have compared with, we can attempt to include such an evaluation.
>
> > RLIF reduced to a heuristic to define a reward function
>
> Our method does not require the expert to define a reward function, but rather to provide interventions. While the way this is used to induce a reward function in RL, that is not a heuristic: we show that this leads to a reward signal that allows our method to recover a policy that is close to the optimal policy (under some assumptions).
>
> > compare with DAgger in RL
>
> We are not sure what "using DAgger in RL" means -- could you provide a reference of a method that does this? The advantage of our method over DAgger is illustrated in the empirical comparisons. For example, Table 1 shows our method is 2-3x better than DAgger.

---

> ### Author Response · Authors · 2023-11-21
> **Friendly reminder**
>
> Dear reviewer,
>
> We greatly appreciate your time and dedication to providing us with your valuable feedback. We hope we have addressed the concerns, but if there is anything else that needs clarification or further discussion, please do not hesitate to let us know.
>
> Best,
> Authors

---

> ### Comment · Reviewer_guRK · 2023-11-21
> **Thanks for your comment**
>
> I apologize for the lack of clarity when I wrote my review. I am concerned with the lack of comparison with the common approach of warm starting the policy training with Dagger and proceeding with PPO. The heuristic determination of when to execute the expert is equivalent to defining an arbitrary reward function, so in my opinion that would be a fair comparison. I would like to know the author's opinion on why this is not the case.
>
> Edit: Regardless of this discussion, which I do consider necessary for this paper, I think my rating is unnecessarily hard, so I'll correct it to a fairer score while waiting for a chance for the authors to respond to my question, which I hope is clearer now.

---

> > ### Author Response · Authors · 2023-11-22
> >
> > Thank you for your reply! now I see your concern better but do correct me if I am still not getting your point!
> >
> > We can't run PPO with the true reward because it's not known, we could run it with the same intervention reward, but this would simply amount to a version of our method that uses PPO instead of RLPD; because our method in principle works with any RL method, and in fact we added a new IQL baseline in the updated draft already.
> > We can definitely try to add this for the final but unfortunately, we are running out of time right now.
> >
> > Please let us know if this addresses your concern or if not, maybe you could further clarify your questions.
> >
> > Thanks,
> > Authors

---

> ### Comment · Reviewer_guRK · 2023-11-22
> **Thanks for the last minute response**
>
> I wish there had been a chance to further discuss what I consider a reasonable question about the usability of the proposal. It's not clear to me that designing a heuristic to call an expert is essentially easier than defining a reward function, but the limited discussion period (and my late receipt of the authors' answer) made it impossible for me to clarify this point.
>
> In any case, I thank the authors for considering to include such an experiment in a potential final version and, since my concern doesn't seem to be relevant in the other reviews, I won't hold a strong opinion about the paper, which otherwise is well written and even shows real-world deployment of learned policies.

---

> > ### Author Response · Authors · 2023-11-22
> >
> > Thank you for the reply.  We are not designing a heuristic in any way. The intervention modes in the simulation experiments are some reasonable ways to model human interventions for which RLIF can use as rewards, so that we can do controlled study, because we need to model human behavior somehow in simulations.
> > For example, the value-based intervention strategy is just one way we think is reasonable for a human to issue an intervention. And in fact, as supported by our theory section, the more task-relevant information contained in such intervention signals (i.e., on which situation a humen actually intervenes), the less the suboptimality gap is.  This is supported by experiment results that value-based (which contains task-relevant information) almost certainly performs better than random intervention (zero task information).  In real world experiment, we actually conducted human study where RLIF has to learn from noisy stochastic real human feedback (still , no ground truth reward), and the results show that RLIF learns very effectively in real-world complex vision-based robotic tasks, and outperform DAgger especially when the suboptimality gap is large.
> >
> > We can't run PPO with ground truth reward because it's not available, what we were saying was that we could run PPO with the intervention reward, which is basically another variant of our method.
> >
> > I hope this can clarify the part about heuristics, and please do let us know if you have further concerns, we would be happy to discuss more.
> >
> > Thanks,
> > Authors

---

> ### Comment · Reviewer_guRK · 2023-11-22
> **Response**
>
> First of all, thank you so much again for your quick response.
>
> Even in the case where the intervention is not heuristic, the reference value function has been learned from data, so a fair comparison might still include using e.g. IRL to learn a reward function and apply e.g. some on-policy algorithm like PPO after a Dagger warm start. I wouldn't want to propose such a costly experiment for the rebuttal, and, at the end of the day, the actual decision of when to call the expert in the proposed method is still a heuristic on top of the learned reference value function, so a heuristic could in principle also be used to define a competent reward function as a comparison, which should easily beat at least the random intervention baseline and would show how far a reasonable but not optimal heuristic reward function is from the one defined with the proposal.
>
> In any case, as I mentioned above, since this seems to be of concern only to me, I won't argue against the paper based on that. Again, thank you for your responses.

---

> > ### Author Response · Authors · 2023-11-22
> > **Clarification**
> >
> > Thank you for the response. We think there is actually a misunderstanding here (that we finally understood from your last comment): our method is not deciding when to call the expert via a heuristic, the expert themselves is deciding when to intervene. The "heuristic" that we think you are referring to (the intervention model in Section 5.1) is simply a computational model of a human that we use in our simulated experiments so that we can run large-scale quantitative comparisons without a real human in the loop. The real-world experiments in Sec 5.3 use a real human expert, who decides when to intervene on their own. No value function or other heuristic is needed to determine when an intervention should take place. This is why we said PPO can only be used with the same intervention reward (which would just amount to our method with PPO instead of RLPD). We agree that IRL can in principle be used, but the "textbook" way of doing this would be to run IRL on the initial set of demos, since IRL methods do not benefit from interventions classically. This would be a much smaller amount of data, and likely perform significantly worse (though we could attempt such an experiment for the final, likely not enough time in the rebuttal period).
> >
> > **Does this clarification resolve your concern?**
> >
> > We apologize for the misunderstanding and will edit the paper to make this point clearer.

---

> > > ### Comment · Reviewer_guRK · 2023-11-22
> > > **Thank you**
> > >
> > > That was finally clearing my concerns about the paper. Thank you so much for the detailed response.

---

> > > > ### Author Response · Authors · 2023-11-22
> > > >
> > > > Great, thank you so much for your time!
> > > >
> > > > If this addresses your concern, would you mind increasing the score?

---

> > > > > ### Comment · Reviewer_guRK · 2023-11-22
> > > > > **Done**
> > > > >
> > > > > I did.

---

> > > > > > ### Author Response · Authors · 2023-11-22
> > > > > >
> > > > > > Thank you!

---

### Official Review · Reviewer_RarS · 2023-11-01

**Soundness:** 3 good
**Presentation:** 3 good
**Contribution:** 3 good
**Rating:** 6
**Confidence:** 3

**Summary:**

This paper introduces a novel off-policy RL method that uses user intervention signals as rewards, allowing learning beyond the limitations of potentially suboptimal human experts. This approach is less reliant on near-optimal expert input compared to DAgger. The authors provide a unified analytical framework for their method and DAgger, including asymptotic and non-asymptotic evaluations. They validate their method with high-dimensional simulations and real-world robotic tasks, showing its practical effectiveness in surpassing traditional imitation learning methods in complex control scenarios.

**Strengths:**

(1) The motivation behind this paper is natural and valuable.

(2) The theoretical analysis is sufficient.

(3) The authors conduct some experiments on real robot arms, which proves the effectiveness of the algorithm.

**Weaknesses:**

(1) The author should include more baselines in Table 1, where the mentioned methods only contain DAgger. For example, some IRL methods could also be applied in the same settings.

(2) Does this method require high frequencies to intervene? Can you do some ablations for this? If RLIF cannot work under high frequent intervenes, it can be hard to apply in more complicated real-world scenarios.

**Questions:**

I am wondering what the difference is between RLHF and your work RLIF when you choose the Space Mouse to give signals by hand. No related works are mentioned and no experiments are conducted for this. For example, what about using some foundation models to give signals?

---

> ### Author Response · Authors · 2023-11-19
>
> Dear Reviewer RarS,
>
> Thank you for your thoughtful feedback, we would like to address your concerns below.
>
> >Regarding baselines:
>
> our aim is to develop a reinforcement learning method that can operate under the same assumptions as interactive imitation learning (e.g., DAgger), hence we compare to DAgger, HG-DAgger, and BC. While IRL methods could in principle be adapted to this problem setting also, this would be an awkward fit, as to our knowledge IRL methods do not use interventions.
> However, we did conduct new experiments using IQL as an offline RL baseline across all simulation benchmarks, and the results are detailed in Table 1.
>
> >Regarding your question about intervention frequencies,
>
> our real-world robot experiments with vision-based peg insertion and deformable object manipulation (in Section 5) already demonstrated the effectiveness of our method under both sparse and dense interventions, i.e., the user intervenes only a few times, but each intervention length is long vs an user intervenes multiple times for a short time during a rollout. In practical terms, the choice between sparse and dense interventions is problem-dependent, varying according to the specific needs of each scenario.
> We have also included a plot of the average intervention rate over environment steps in Figure 3, illustrating that at the onset of training, the intervention rate tends to be higher, gradually decreasing as the policy improves towards the end of the training process.
>
> >RLIF vs RLHF:
>
> Thank you for your suggestion! RLHF indeed can be relevant to our work, we’ll add some related works in the final version.
>
> Your engagement with our work is highly valued, and we welcome any further questions or insights you may have.
>
>
> Table 1: IQL Baseline
>
> |                        | IQL with Random Interventions | IQL with Value Based Interventions |
> |------------------------|------------------------------|------------------------------------|
> | **adroit-pen**         |                              |                                    |
> | Expert Levels          | 68.83                        | 42.5                               |
> | 90% Expert             | 25.92                        | 46.58                              |
> | 40% Expert             | 12.08                        | 40.67                              |
> | 8% Expert              | 35.61                        | 43.25                              |
> |                        | 80.22                        | 20.37                              |
> | **locomotion-walker2d**|                              |                                    |
> | 110 Expert             | 56.33                        | 47.97                              |
> | 70 Expert              | 14.7                         | 47.02                              |
> | 20 Expert              | 50.42                        | 38.45                              |
> |                        | 37.91                        | 26.37                              |
> | **locomotion-hopper**   |                              |                                    |
> | 110 Expert             | 27.88                        | 28.98                              |
> | 40 Expert              | 16.64                        | 25.32                              |
> | 15 Expert              | 27                           | 26.89                              |

---

> ### Author Response · Authors · 2023-11-21
> **Friendly reminder**
>
> Dear reviewer,
>
> We greatly appreciate your time and dedication to providing us with your valuable feedback. We hope we have addressed the concerns, but if there is anything else that needs clarification or further discussion, please do not hesitate to let us know.
>
> Best,
> Authors

---

> > ### Author Response · Authors · 2023-11-22
> > **Deadline approaching, friendly reminder**
> >
> > Dear Reviewer RarS,
> >
> > Since the rebuttal deadline is approaching in 24 hours, we would like to send another friendly remind.
> >
> > We have updated our draft with changes marked as red, you could view it by downloading the new PDF.
> >
> > Specifically, we have also addressed your concern about IRL as well as intervention frequencies.
> >
> > Would you kindly let us know if this addresses your concerns and if not, we'd happy to address your remaining questions.
> >
> > Best,
> > Authors

---

### Official Review · Reviewer_9AYW · 2023-11-04

**Soundness:** 3 good
**Presentation:** 3 good
**Contribution:** 3 good
**Rating:** 6
**Confidence:** 4

**Summary:**

This paper tackles how to enable sub-optimal human interventions (sub-expert samples) to improve a RL agent. It falls in two sub-domains in the field: interactive imitation learning, and reinforcement learning from human feedback.

The authors explore the assumption that the decision to intervene provides an effective supervision signal. Accordingly, a solution based on augmentation the reward with intervention penalty, and off-policy learning is used to minimize such a penalty (maximize the accumulative negative reward), so that the RL agent achieve 3X performance gain compared to DAgger, and BC (bahevior cloning) baselines. Ablation studies on sub-optimality of expert samples are also provided.

**Strengths:**

Novelty:
========
The assumption of “when an sub-optimal expert to intervene” provides useful learning signals is novel to me.

Soundness:
========
Accordingly, the authors provide both empirical results in simulation and real-world robotic task, and theoretical justifications by proposing a probabilistic model of “when human expert will intervene”.

**Weaknesses:**

- Though the theoretical analysis does not answer the key problem, how the level of sub-optimality results in what sample complexity of the offline RL agent. The reviewer still enjoys reading the analysis framework proposed here. The reviewer may miss some key contents, but would the authors explain more in the proposed Corollary 6.7 regarding the above key problem?
- Baselines:
There is no offline-RL baselines provided. Is it because that the true reward function is unknown in the evaluation results? Otherwise, please add at least 1 offlineRL baseline in all the simulation based experiments.

**Questions:**

Please check the two questions the reviewer proposed in the Weakness part.

---

> ### Author Response · Authors · 2023-11-19
>
> Dear Reviewer 9AYW,
>
> Thanks for your insightful feedback on our work, we would like to address your concerns below.
>
> To answer your question regarding the theoretical results:
>
> > how the level of sub-optimality results in what sample complexity of the offline RL agent. The reviewer still enjoys reading the analysis framework proposed here.
>
> Corollary 6.7 provides a sample complexity bound of learning a policy using the intervention reward $\tilde{r}$. Since the intervention strategy is generated by the expert $\pi^{\exp}$, which eventually affects the suboptimality gap in a linear relationship w.r.t. $\Delta_{ref}$, the visitation distribution gap between the reference policy $\pi^{ref}$ and the optimal policy $\pi^{\star}$.
>
> > Concerning your question about offline RL baselines
>
> First, RLIF in principle can be integrated with any off-policy RL algorithm. We chose RLPD because of its good sample efficiency. That said, per your suggestion, we conducted additional experiments incorporating IQL as an offline RL baseline across all simulation benchmarks. Please find the detailed results in Table 1. It's worth noting that, in all our experiments, true rewards were not provided, as our algorithm is designed to learn an optimal policy with only intervention feedback.
>
> Upon examination, the results from the IQL experiments indicate a performance that lags behind the outcomes achieved using the online algorithm RLPD. This observation aligns with our expectations because it’s slow for IQL to explore new actions.
>
> Thank you for your continued engagement with our work, and we remain open to any further questions or suggestions you may have.
>
> Table 1: IQL Baseline
>
> |                        | IQL with Random Interventions | IQL with Value Based Interventions |
> |------------------------|------------------------------|------------------------------------|
> | **adroit-pen**         |                              |                                    |
> | Expert Levels          | 68.83                        | 42.5                               |
> | 90% Expert             | 25.92                        | 46.58                              |
> | 40% Expert             | 12.08                        | 40.67                              |
> | 8% Expert              | 35.61                        | 43.25                              |
> |                        | 80.22                        | 20.37                              |
> | **locomotion-walker2d**|                              |                                    |
> | 110 Expert             | 56.33                        | 47.97                              |
> | 70 Expert              | 14.7                         | 47.02                              |
> | 20 Expert              | 50.42                        | 38.45                              |
> |                        | 37.91                        | 26.37                              |
> | **locomotion-hopper**   |                              |                                    |
> | 110 Expert             | 27.88                        | 28.98                              |
> | 40 Expert              | 16.64                        | 25.32                              |
> | 15 Expert              | 27                           | 26.89                              |

---

> ### Author Response · Authors · 2023-11-21
> **Friendly reminder**
>
> Dear reviewer,
>
> We greatly appreciate your time and dedication to providing us with your valuable feedback. We hope we have addressed the concerns, but if there is anything else that needs clarification or further discussion, please do not hesitate to let us know.
>
> Best,
> Authors

---

> > ### Author Response · Authors · 2023-11-22
> > **Deadline approaching, friendly reminder**
> >
> > Dear reviewer 9AYW,
> >
> > Since the rebuttal deadline is approaching in 24 hours,  we would like to send another friendly remind.
> >
> > We have updated our draft with changes marked as red, you could view it by downloading the new PDF.
> >
> > Specifically, we have also addressed your concern by adding an IQL baseline, as well as having an updated theory section, and you are right, the  Corollary 6.7 in the old manuscript does address the sample complexity w.r.t. the suboptimality gap.
> >
> > Would you kindly let us know if this addresses your concerns and if not, we'd happy to address your remaining questions.
> >
> > Thanks,
> > Authors

---

### Author Response · Authors · 2023-11-19
**Summary of changes**

Dear reviewers,

Thank you for your comments in helping us strengthen our paper, we have incorporated many of your suggestions, and we uploaded a new PDF containing revisions,  the changes were marked as red, you could download it from the PDF button in the top of this page.  We summarize our main changes below:
* Added IQL as an offline RL baseline, which is included in the Appendix A.6
* Added more challenging real robot experiments: inserting two-prog pegs into the matching hole, as well as unfolding a deformable cloth. These changes can be found in Section 5.3
* Improved the theory section (6.2) where we show the DAgger upper bound under our framework is tight, and a better RLIF regret bound which is at least as good as DAgger. More details can be found in Appendix B and C.
* We have updated our website: https://rlifpaper.github.io/ to include more real-robot videos.

Please let us know if you have any other concerns or need clarification of the changes we made.

Authors

---

### Public Comment · ~Zhenghai_Xue1 · 2023-11-27
**Suggestions on Missing References of Interactive Imitation Learning**

Dear Authors of Submission 5065,

Thank you for presenting this exciting work of human-in-the-loop training of robot manipulation. However, I would suggest considering the following three papers that are very relevant to this paper:

[1] [Guarded Policy Optimization with Imperfect Online Demonstrations](https://arxiv.org/abs/2303.01728). ICLR 2023.

[2] [Efficient Learning of Safe Driving Policy via Human-AI Copilot Optimization](https://arxiv.org/abs/2202.10341). ICLR 2022.

[3] [Safe Driving via Expert Guided Policy Optimization](https://arxiv.org/abs/2110.06831). CoRL 2022.

Similar to this paper, paper [1] focuses on the setting of interactive decision-making without an optimal expert. The intervention function and the theoretical results (Eq. 5 and Thm. 3.4) in [1] are very close to those in this paper (Eq. 5.1 and Thm. 6.3). Also similar to this paper, paper [2] focuses on human-in-the-loop decision-making without task reward assumptions. With the help of human drivers, the HACO algorithm in Paper [2] enables safe and efficient training of autonomous driving agents. Paper [3] focuces on the foundamental formulation of interactive imitation learning and motivates Paper [1,2]. So it may also be considered.

I will be appreciated if the authors of submission 5065 can kindly consider these three papers as reference in the next version of this paper. Thanks for your time.

Best Regards,

Zhenghai Xue

---

### Meta-Review · Area_Chair_9zSy · 2023-12-06

**Metareview:**

The paper proposes a new off-policy RL method that uses user intervention signals as rewards, allowing learning beyond the limitations of potentially suboptimal human experts. The paper has strong and natural motivations, and sufficient experiments both in simulations and on real-robotic platforms. It also provides some theoretical justifications for the proposed approach. Overall the paper is well-written, and it reaches a consensus that this is a good paper and should be accepted.

**Justification For Why Not Higher Score:**

Though the overall framework is novel, some experiments could have been stronger by including more baselines and ablation studies. The theoretical results are mostly based on existing techniques in RL theory, and not extremely strong. The oversight of some related literature (and comparison with them) could have strengthened the paper further.

**Justification For Why Not Lower Score:**

It is a good paper that can potentially motivate more research along the line, and it is well-written, with good experimental and theoretical results.

---

### Decision · Program_Chairs · 2024-01-16

Accept (poster)